

# Multi-year CO₂ budgets in South African semi-arid Karoo ecosystems under different grazing intensities

Oksana Rybchak[1], Justin du Toit[2], Jean-Pierre Delorme[1], Jens-Kristian Jüdt[1], Kanisios Mukwashi[1], Christian Thau[3], Gregor Feig[4,5], Mari Bieri[1], Christian Brümmer[1]

[1]Thünen Institute of Climate-Smart Agriculture, Braunschweig, 38116, Germany
[2]Grootfontein Agricultural Development Institute, Middelburg, 5900, South Africa
[3]Department of Earth Observation, Friedrich-Schiller University Jena, Grietgasse 6, 07743 Jena, Germany
[4]South African Environmental Observation Network, Colbyn, Pretoria, 0083, South Africa
[5]Department of Geography, Geoinformatics and Meteorology, University of Pretoria, Pretoria, 0002, South Africa

*Correspondence to*: Oksana Rybchak (oksana.rybchak@thuenen.de), Christian Brümmer (christian.bruemmer@thuenen.de)

**Abstract.** Climatic and land management factors, such as water availability and grazing intensity, play an important role in seasonal and annual variability of the ecosystem–atmosphere exchange of $CO_2$ in semi-arid ecosystems. However, the semi-arid South African ecosystems have been poorly studied. Four years of measurements (November 2015–October 2019) were collected and analysed from two eddy covariance towers near Middelburg in the Karoo, Eastern Cape, South Africa. We studied the impact of grazing intensity on the $CO_2$ exchange by comparing seasonal and interannual $CO_2$ fluxes for two sites with almost identical climatic conditions but different intensity of current and historical livestock grazing. The first site represents lenient grazing (LG) and the vegetation comprises a diverse balance of dwarf shrubs and grasses, while the second site has been degraded through heavy grazing (HG) in the past but then rested for the past 10 years and mainly consists of unpalatable grasses and ephemeral species. Over the observation period, we found that the LG site was a considerable carbon source (82.11 g C m$^{-2}$), while the HG site was a slight carbon sink (–36.43 g C m$^{-2}$). The annual carbon budgets ranged from $-90 \pm 51$ g C m$^{-2}$ yr$^{-1}$ to $84 \pm 43$ g C m$^{-2}$ yr$^{-1}$ for the LG site and from $-92 \pm 66$ g C m$^{-2}$ yr$^{-1}$ to $59 \pm 46$ g C m$^{-2}$ yr$^{-1}$ for the heavily grazed site over the four years of eddy covariance measurements. The significant variation in carbon sequestration rates between the last two years of measurement was explained by water availability (25 % of the precipitation deficit in 2019 compared to the long-term mean precipitation). This indicates that studied ecosystems can quickly switch from a considerable carbon sink to a considerable carbon source ecosystem. Our study shows that the $CO_2$ dynamics in the Karoo are largely driven by water availability and the current and historical effects of livestock grazing intensity on aboveground biomass (AGB). The higher carbon uptake at the HG site indicates that resting period after overgrazing, together with the transition to unpalatable drought-tolerant grass species, creates conditions that are favourable for carbon sequestration in the Karoo ecosystems, but unproductive as Dorper sheep pasture. Furthermore, we observed a slight decrease in carbon uptake peaks at the HG site in response to resuming continuous grazing (July 2017).



## 1 Introduction

Global environmental changes (such as global warming, land degradation, etc.) alter ecosystem processes such as photosynthesis, respiration and decomposition, therefore affecting the carbon cycle (Cao et al., 2001; Groendahl et al., 2007; Jung et al., 2017; Miranda et al., 2011; Müller et al., 2007; Yi et al., 2012). Assessment of carbon uptake or release has turned into one of the most critical issues in environmental science in the last decade (Araújo et al., 2002; Ardö et al., 2008; K'Otuto et al., 2013; Nakano and Shinoda, 2018). Even though numerous studies on ecosystem carbon budget have been conducted, most of them have concentrated on European, Asian or North American mid-latitude ecosystems (Anthoni et al., 2004; Bao et al., 2019; Béziat et al., 2009; Fei et al., 2017; Nagy et al., 2006; Niu et al., 2020; Rannik et al., 2002; Rogiers, 2005; Schmid, 2002; Wang et al., 2016; Yan et al., 2017), while a lack of attention has been given to the African continent. Even though South Africa has been heavily affected by climate change during the recent decades (Hulme et al., 2001; Ziervogel et al., 2014), many gaps remain in the understanding of the semi-arid ecosystem carbon exchange (Archibald et al., 2009; Kutsch et al., 2008; Räsänen et al., 2017; Talore et al., 2016; Veenendaal et al., 2004).

Semi-arid ecosystems contribute up to 20 % of terrestrial net primary productivity and are thus of particular concern for the global carbon balance (Ahlström et al., 2015; Meza et al., 2018). By partitioning terrestrial $CO_2$ among land cover classes, Ahlström et al. (2015) showed that the interannual variability of the net biome production is dominated (39 %) by semi-arid ecosystems. The strong diurnal, seasonal and interannual variability in $CO_2$ fluxes indicates that carbon exchange can be affected by the ecosystem's response to land management and meteorological forcing (Archibald et al., 2009; Brümmer et al., 2008; Merbold et al., 2009; Nakano and Shinoda, 2018; Räsänen et al., 2016; Sorokin et al., 2017).

Previous studies determined water availability as the main driver controlling gross primary production (GPP) through increasing photosynthesis and extending the length of the growing season (Ago et al., 2016; Bao et al., 2019; Meza et al., 2018; Otieno et al., 2010; Wang et al., 2019). However, the seasonal responses of GPP and ecosystem respiration ($R_{eco}$) to precipitation pattern and the impacts of precipitation on the annual carbon budget are less well understood in the South African semi-arid ecosystems. These ecosystems are highly seasonal with large parts of total annual precipitation occurring in only a few months of the year, and generally reveal pulse-driven evapotranspiration events in the rainy season. Phase and amplitude of photosynthesis and respiration are also highly dependent on the precipitation pattern (Archibald et al., 2009; Bao et al., 2019; Kutsch et al., 2008; Merbold et al., 2009; Räsänen et al., 2017). The strength of the ecosystem response to rainfall is not only determined by the amount of an individual event, but also on the timing of preceding events (Huxman et al., 2004; Ivans et al., 2006; Veenendaal et al., 2004; Williams et al., 2009; Yepez and Williams, 2009).

Land and vegetation degradation belong to the crucial drivers altering ecosystem–atmosphere carbon exchange. Even though many studies suggest that overgrazing leads to biodiversity losses and soil degradation (Hansis et al., 2015; Kairis et al., 2015), its impact on the ecosystem $CO_2$ exchange is still not well understood as ecosystems may respond differently depending on their grazing history as well as biotic (vegetation cover) and micrometeorological factors (Chen et al., 2014; Gourlez de la Motte et al., 2018; Müller et al., 2007; Richter and Houghton, 2011; Talore et al., 2016; du Toit et al., 2011; Wang et al., 2019).





65    Most studies providing comparisons of carbon exchange under different grazing intensities are of short duration, hence they do not detect seasonal nor annual changes in the carbon dynamics associated with grazing, and are not based on South African semi-arid ecosystems (Chen et al., 2014; Chimner and Welker, 2011; Frank, 2002; Lecain et al., 2000; Wang et al., 2016; Yan et al., 2017). Assessing the impacts of livestock grazing on the ecosystem carbon exchange and vegetation cover is important for understanding the driving factors of carbon dynamics in the semi-arid grazed ecosystems. According to Reyers et al. (2009),

70    about 52 % of the semi-arid Karoo has been moderately or severely degraded due to livestock overgrazing. However, Hoffman et al. 2018 showed that the situation has improved somewhat at Nama-Karoo for example, by reduction in the number of sheep from 11 million (1939) to 4 million (2007), and by extension of the protected area (from 0.03 % to 1.6 %).

In this study, we investigated the combined impacts of livestock grazing and the interannual climatic (precipitation) variability on the ecosystem–atmosphere $CO_2$ exchange. The study is based on four years of continuous eddy covariance (EC)

75    measurements of $CO_2$ at semi-arid Karoo near Middelburg, Eastern Cape, South Africa. Our paired-tower approach allowed for an analysis of the effects of different grazing regimes on $CO_2$ fluxes under identical climate. The two study sites represent lenient temporary (LG) vs. continuous heavy grazing (HG). We hypothesize that the HG regime reduces the ecosystem carbon sink potential by altering vegetation cover, decreasing above-ground biomass (AGB) and gross primary production. In our analysis we attribute the difference in $CO_2$ exchange between the two sites to current and historic grazing intensity, while

80    changes in yearly $CO_2$ budgets are mainly explained by inter-annual variability of rainfall sums and distribution.

## 2 Methodology

### 2.1 Sites description

The two study sites were located at an altitude of 1310 m.a.s.l, in the Eastern Upper Karoo vegetation type in the Nama-Karoo, a biome that occupies much of the central to western inland region of South Africa (Mucina et al., 2006) (Fig. 1). The

85    vegetation is co-dominated by dwarf shrubs (perennial, both succulent and non-succulent) and grasses (short-lived and perennial), with shrubs, geophytes, sedges and herbs also present. The soils are loamy at both study sites (Roux, 1993; du Toit and O'Connor, 2020). The climate is characterised by long hot summers and moderate to warm winters (du Toit and O'Connor, 2014). During the summer months, days are generally hot (30–40 °C) and nights are moderately warm (10–16 °C), while the winter days are moderate to warm (14–25 °C) and nights are cold (−4–4 °C). The long-term mean annual temperature was 15

90    °C. Cool seasons (April–September) and warm seasons (October–March) can be distinguished throughout all years. Mean annual precipitation was 374 mm from 1889 to 2013 in the range from 163 mm to 749 mm (du Toit et al., 2015).





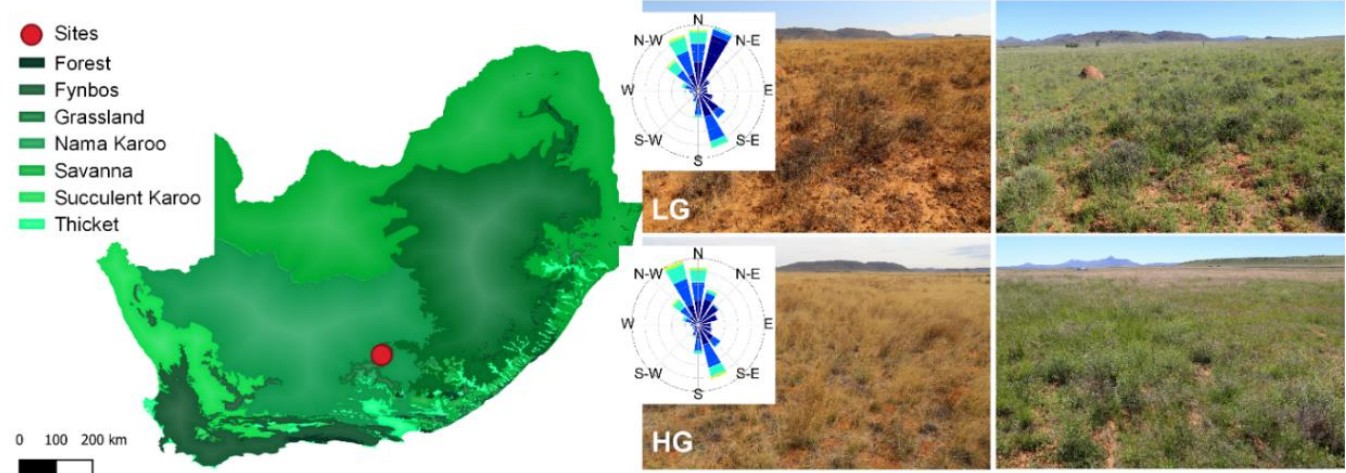

**Figure 1.** South African biomes and location of the studied sites marked as a red circle with footprints, wind roses on the right side and pictures in dry and growing seasons for the lenient grazing (LG) site (top) and the heavy grazing (HG) site (bottom) (Department of Agriculture Fisheries and Forestry, 2013).

The Lenient Grazing (LG) (31°25'20.97" S, 25°1'46.38" E) site has been grazed by sheep and cattle using a rotational grazing system (approximately 2 weeks grazing followed by 24–26 weeks rest) at recommended stocking rates of 16 hectares per animal unit (ha AU$^{-1}$) since the 1970s, and is considered to be an excellent condition 'benchmark' site in terms of botanical composition, with a wide diversity of species and co-dominance of grasses and dwarf-shrubs (du Toit, 2002). Dominant species are *Digitaria eriantha* (palatable perennial grass) and *Pentzia globosa* (palatable non-succulent dwarf-shrub) (du Toit and Nengwenani, 2019). The Heavy Grazing (HG) (31°25'48.69''S, 25°0'57.70''E) site was grazed by Dorper sheep using a 2-paddock rotational grazing system (120 days grazing followed by 120 days rest) at stocking rates approximately double that of the recommended rate as part of an experimental trial from 1988 to 2007. The Dorper breed is described as a hardy sheep that prefers shrubs to a greater extent than grasses (Brand, 2000). The site was ungrazed 2008–2017 but did not recover, after which Dorpers were reintroduced at a similar stocking rate in July 2017, and the veld was grazed continuously unless (for short periods) food availability was too low. This severe treatment extirpated nearly all palatable species and nearly all dwarf shrubs, and as a result, the system is dominated by *Aristida diffusa* (unpalatable perennial grass) and *Aristida congesta* (short-lived unproductive grass) (van Lingen, 2018). Thus, the HG site is a degraded site from an agricultural point of view, having shifted from a diverse grassy shrubland to unpalatable arid grassland. Climatic conditions of the two sites are similar. There is no information on the difference between the sites in bulk density, soil nitrogen and organic carbon content.

## 2.2 Instrumentation and measurements

Eddy Covariance (EC) towers were installed at the LG and HG sites to measure ecosystem–atmosphere exchange of carbon, water and heat fluxes in October 2015. The two towers are placed approximately 1.5 km apart. Wind components were measured with a three-dimensional sonic anemometer (CSAT3, Campbell Scientific Inc., Logan, UT, USA) placed 3 m above



the ground. The sonic anemometer was coupled with an enclosed path fast-response Infra-Red Gas Analyser (IRGA) Li-7200
(IRGA, Li-Cor, Lincoln, NE, USA) for $CO_2$ and $H_2O$ measurements. Air from close proximity (~5 cm) to the sonic
anemometer's array was pulled through a sampling tube into the IRGA through a stainless steel tube of 70 cm length and 6.0
mm inner diameter using the pump from Licor's Li-7200-101 Flow Module with a flow rate of 15 L min$^{-1}$. The IRGA
acquisition system collected raw data in real-time at 20 Hz sampling rate, compressed files (meta, flux and biomet data) for
later processing. The gas analysers were manually calibrated every six months using standard air calibration tanks with mixing
ratios of zero and span (431.23 ppm). Drift of the sensors was always below 1%.

Relative humidity and air temperature were recorded with Temperature/Humidity Probe (HMP155, Vaisala, Helsinki,
Finland), precipitation with Tipping Bucket Rain Gauge (TR 525, Texas Electronics, Texas, USA), radiation components with
Net Radiometer (CNR4, Kipp&Zonen, Delft, Netherlands). Soil measurements consisted of heat flux plates (HFP01 + HFP01
SC, Campbell scientific, Utah, USA) at two different depths (10 and 20 cm) with three repetitions, soil temperature probes
(UMS TH3 sdi12, Meter AG, Munich, Germany) at six different depths (5, 6, 10, 13, 20 and 37 cm at LG site and at 10, 12,
17, 22, 32 and 57 cm at HG site), soil moisture probe (ML3x Delta T, EcoTech, Bonn, Germany) at two different depths (10
and 16 cm at the LG site and at 8 and 14 cm at the HG site). The two sites differed in soil heterogeneity and characteristics
particularly in stone content (soil skeleton >2 mm for the HG site), which were the reasons for different installation depths of
the soil temperature and moisture probes.

Also, with the EC installation, we were able to evaluate latent and sensible heat fluxes, energy balance closure, and diurnal
variation of the residual of the energy balance. Figures dealing with energy flux topics are briefly described in the Appendix
A (Figs. A1, A2, A3, Table A1).

## 2.3 Data processing

### 2.3.1 Eddy covariance post-field data processing

Fluxes were calculated on the basis of the EC technique from raw high-frequency data with a half-hourly averaging period,
discussed and explained in detail in previous studies (Aubinet et al., 2000; Baldocchi et al., 2001; Burba, 2013; Foken and
Wichura, 1996; Moncrieff et al., 1997; Webb et al., 1980) strictly following the recently published processing chain of the
FLUXNET methodology by Pastorello et al. (2020). Processing was performed with the EddyPro software package version
7.0.6 to quantify ecosystem–atmosphere exchange of $CO_2$, water vapour, and sensible heat fluxes. Flux data were available
for the four hydrological years from November 2015 to October 2019. Complete datasets consist of 70,129 30-min flux
measurements with 8 % for LG site and 12 % for HG site missing data due to instrument failure. After processing, the gaps
increased to 12 % and 26 % of the missing data for LG and HG sites, correspondingly.

The absolute limits assessment was performed to filter out implausible data by setting upper and lower thresholds (Aubinet et
al., 2000; Baldocchi, 2003; Foken and Wichura, 1996). Spikes were removed by applying the median absolute deviation
(MAD) filter (Mauder et al., 2013). The amplitude resolution test to identify data with a low variance was performed according





to Vickers and Mahrt (1997). The vertical and horizontal velocity components were transformed to zero to adjust wind statistics for the misalignment of the sonic anemometer by performing a two-dimensional coordinate system rotation (Aubinet et al., 2000; Baldocchi et al., 1988; Wilczak et al., 2001). The time lag compensation between the anemometric and gas analyser variables was detected using the maximum covariance method (Fan et al., 1990). Linear detrending was used to extract turbulent fluctuations from time-series data (Gash and Culf, 1996). Also, fluxes were corrected following systematic losses in

the low and high frequency range due to the data processing choices, system characteristics and sampling ranges of the instruments (Moncrieff et al., 1997, 2006). Quality control flags were calculated for all fluxes by the flagging methodology after Foken et al. (2006) and Mauder and Foken (2011). After application of all the correction procedures, half-hourly values of $CO_2$, latent and sensible heat fluxes were calculated.

### 2.3.2 Flux filtering and quality control

Based on the criteria after Mauder and Foken (2011), bad quality flux data that obtained the value "2" were discarded from further analyses. In total, after the quality checking and quality flags filtering procedures, the missing $CO_2$ flux measurements data increased to 27 % for the LG site and 34 % for the HG site.

Numerous studies indicate reduced reliability of the nighttime flux measurements due to periods with low turbulent mixing of the planetary boundary layer (Aubinet et al., 2000; Gu et al., 2005; Twine et al., 2000). Many authors describe a threshold

value for the friction velocity ($u_*$) filtering criterion to remove periods of intermittent turbulence as the EC assumptions fail during low turbulence periods (Aubinet et al., 2000, 2012; Baldocchi, 2003; Goulden et al., 1996). To identify insufficient turbulent mixing, we estimated the minimum friction velocity $u_*$ according to the method described in Papale et al. (2006). Threshold values of $u_*$ were 0.10 m s$^{-1}$ for both sites. The corrected fluxes were filtered by removing data below these thresholds from the dataset. After $u_*$ filtering application, the missing $CO_2$ fluxes data increased to 44 % and 49 % for the LG

and HG site, respectively.

After the standard quality assessment and quality control, flux data sets were again checked and filtered for implausible values. There were a few longer periods of setup malfunction, most of which related to solar panel outages and battery or pump failures, that lasted for 31 days (10/06/2016–10/07/2016) and 30 days (02/01/2018–31/01/2018) on the LG site and for 33 days (23/11/2017–15/12/2017) and 104 days (03/02/2018–17/05/2018) on the HG site.

### 175 2.3.3 Data gap-filling and flux partitioning

The gaps due to the precipitation, non-turbulence conditions, spikes and power breakdowns resulted in 29,641 and 35,942 of 30-min flux gaps for LG and HG sites, respectively. These gaps were filled to obtain continuous dataset for the calculations of the net $CO_2$ balance and its component fluxes GPP and $R_{eco}$.

$CO_2$ fluxes were gap-filled and GPP and $R_{eco}$ were derived using the gap-filling and flux partitioning REddyProcWeb online

tool. The detailed explanation of the gap-filling and partitioning methods are described by Wutzler et al. (2018). The marginal distribution sampling (MDS) method after Reichstein et al., (2005) is implemented into the tool, which integrates the

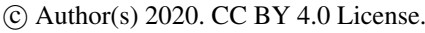



covariation of the fluxes with the micrometeorological conditions within a moving time window and the temporal fluxes' autocorrelation based on look-up tables (LUT) and the mean diurnal course (MDC) method (Falge et al., 2001; Wutzler et al., 2018).

The main components of the carbon fluxes (GPP and $R_{eco}$) can be derived by two methods: the so-called nighttime approach by Reichstein et al. (2005) and the daytime approach by Lasslop et al. (2010). Both approaches were tested extensively for ecosystems mainly in the mid-latitudes but less for highly seasonal ecosystems. In the current study, we used the nighttime approach from Reichstein et al. (2005) with the algorithm fitting a respiration model to the measured nighttime $CO_2$ fluxes data and extrapolating it to daytime data using the corresponding temperature measurements.

### 2.3.4 Uncertainty estimation


Flux measurements by EC method are subject to errors, and estimating them is an important criterion for analysing annual net ecosystem exchange (NEE) balances. The annual NEE uncertainty was calculated by taking into account the most significant possible error sources (Finkelstein and Sims, 2001; Lucas-Moffat et al., 2018; Massman and Lee, 2002; Moffat et al., 2007; Richardson et al., 2012): (1) systematic errors associated with advection, flux divergence and tilt correction, (2) random errors

associated with inadequate sample size, and (3) bias errors due to the gap-filling of the unavoidable gaps in the EC data sets. The NEE uncertainty estimation equations are described in detail by Moffat et al. (2007) and Lucas-Moffat et al. (2018).

The bias errors resulting from gap-filling of the flux measurement data is provided by the formula:

$$\delta ASum = Np \cdot BE \ , \tag{1}$$

where $\delta ASum$ is the offset on the annual sum, Np is the number of gap-filled days and BE is the bias error. Moffat et al. (2007)

demonstrate that the effect of different gap-filling techniques on the bias error of the annual balance of NEE for their selection of sites was less than 0.25 g C $m^{-2}$ $d^{-1}$.

The random error (*RE*) was calculated for each 30 minutes flux measurements by EddyPro software using Finkelstein et al. (2001) method. The random error of the annual sum ($\varepsilon ASum$) is calculated by the formula:

$$\varepsilon ASum = \sqrt{\Sigma RE} \ , \tag{2}$$

Both equations were summed up to estimate total NEE uncertainty for the year:

$$NEEtotal\_uncertainty = \Delta ASum + \varepsilon ASum \ . \tag{3}$$

### 2.4 Remote sensing-based NDVI index

Remote sensing-based vegetation indices (VIs) were used to put the observed flux tower measurements into the context of plant functioning and responses to environmental conditions and management. To this end, Moderate Resolution Imaging

Spectroradiometer (MODIS) time series provided by the Terra and Aqua satellites were acquired for the period under investigation. More specifically, the MODIS VI products MOD13Q1 (Terra) and MYD13Q1 (Aqua) were downloaded from the National Aeronautics and Space Administration (NASA) Distributed Active Archive Centre (DAAC,



https://ladsweb.modaps.eosdis.nasa.gov/search/). Subsequently, the VI values contained in these products were extracted for the two flux tower locations and compared to the EC measurements.

Both MOD13Q1 and MYD13Q1 are global data sets featuring a spatial resolution of 250 m and a temporal resolution of 16 days. The VIs that come with the product as a normalized difference vegetation index (NDVI). The NDVI allows for consistent spatio-temporal comparisons of vegetation canopy greenness and canopy structure. The most recent version (collection 6) of MOD13Q1 and MYD13Q1 was employed in this study. For more information, the reader is referred to the products' user guide (Didan et al., 2015) and their algorithm theoretical basis document (ATBD) (Huete et al., 1999).

## 220 3. Results

### 3.1. Meteorological conditions

The measured daily and seasonal variability in the main climatic conditions (air temperature, radiation, relative humidity) were typical for the Nama Karoo biome (Fig. 2). We analysed wet (January–May) and dry (June–December) seasons and defined the following periods as hydrological years (HY):

-    Year I (01/11/2015 – 31/10/2016),

   -    Year II (01/11/2016 – 31/10/2017),

   -    Year III (01/11/2017 – 31/10/2018),

   -    Year IV (01/11/2018 – 31/10/2019).

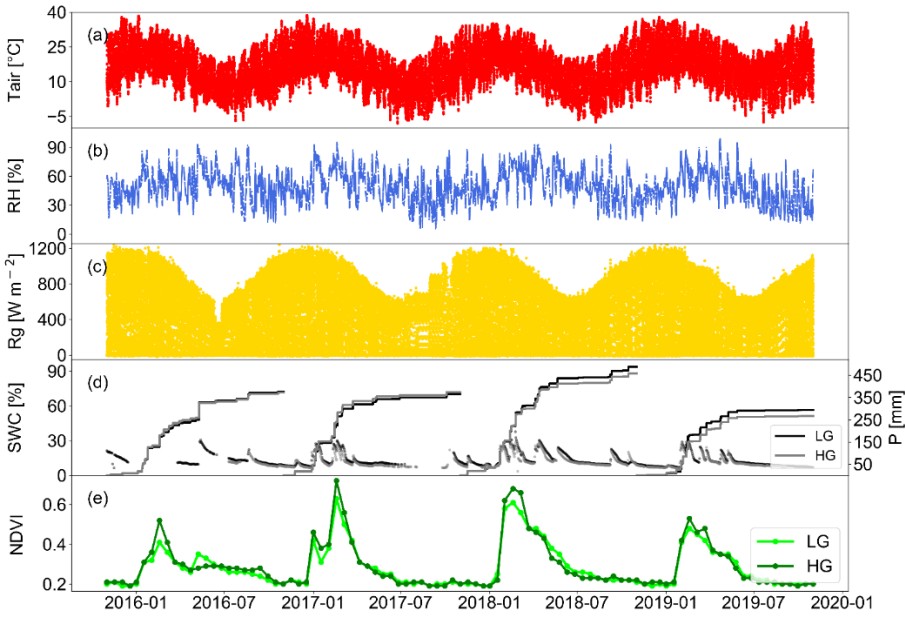

**Figure 2.** Annual and seasonal variations of the micrometeorological variables for the entire measurement period: (a) hourly means of air temperature (Tair), (b) daily means of relative humidity (RH), (c) hourly means of global radiation (Rg), (d) daily means of soil water content (SWC, left) and cumulative precipitation (P, right), (e) normalised difference vegetation index (NDVI) produced on 16-day intervals.





Mean annual temperature (Tair) during the study period (November 2015–October 2019) was 15.7°C for both sites (Fig. 2a, Table 1). Daily mean relative humidity ranged from a minimum of 6% in dry seasons up to 99 % during the growing seasons

(Fig. 2b). Relative humidity (RH) was lowest in November and December, while the highest values were measured in February–May. Short wave incoming radiation ($R_g$) demonstrated typical seasonal fluctuations with higher values during the summer months (Fig. 2c, Table 1). The seasonal changes in $R_g$ showed that the HG site had slightly higher values. The mean daily $R_g$ during the four years measurement period were 240 and 250 W m$^{-2}$ for the LG and HG sites, correspondingly. The peak values reached 1000–1200 W m$^{-2}$ in summer and 400–600 W m$^{-2}$ in winter.


**Table 1.** Summary of meteorological parameters: means of air temperature ($T_{air}$), soil water content (SWC), short-wave incoming radiation (Rg) and cumulative precipitation (P) in the growing seasons (January–May) and dry (June–December). Years I–IV defined as hydrological year (November–October).

| | | $T_{air}$ (°C) | | SWC (%) | | P (mm) | | $R_g$ (W m$^{-2}$) | |
|---|---|---|---|---|---|---|---|---|---|
| | | LG | HG | LG | HG | LG | HG | LG | HG |
| Year I | growing season | 16.7 | 16.9 | 0.151 | 0.101 | 318.5 | 316.1 | 237 | 249 |
| | dry season | 14.7 | 15 | 0.128 | 0.105 | 47.7 | 46.2 | 226 | 245 |
| Year II | growing season | 16.7 | 16.8 | 0.143 | 0.116 | 296.6 | 306.8 | 234 | 248 |
| | dry season | 14.2 | 14.9 | 0.122 | 0.104 | 68.3 | 67.4 | 268 | 255 |
| Year III | growing season | 16.1 | 15.9 | 0.176 | 0.117 | 401.9 | 377.8 | 248 | 251 |
| | dry season | 14.3 | 14 | 0.126 | 0.149 | 85.6 | 80.4 | 236 | 251 |
| Year IV | growing season | 16.1 | 14 | 0.138 | 0.119 | 286.3 | 257.7 | 238 | 242 |
| | dry season | 14.3 | 17 | 0.097 | 0.093 | 8.2 | 9.0 | 245 | 250 |

The LG and HG sites had mean annual precipitation of 378 mm and 365 mm, respectively. Approximately 55% of the total annual precipitation happened during the summer months (January–February), with autumn (March–May) as the second most humid season (approx. 30 %), winter (June–August) as the driest season with precipitation less than 10 mm per month and spring (September–November) characterized by short rain events (Fig. 2c, Table 1). Precipitation was highly variable during the measurement period. Years I and II were nearly the same as the long-term mean annual precipitation for the Karoo (376

mm and 365 mm in the LG site and 372 mm and 374 mm in the HG site) (du Toit et al., 2014). The last year of measurements was the driest year with annual precipitation rates of 295 and 267 mm for LG and HG sites, respectively, while Year III had the highest precipitation of 487 and 458 mm.

Seasonal variation of soil water content (SWC), expressed as volumetric water content, changed with the highest values in the summer and autumn months (growing season) and the lowest values during the winter (Fig. 2d, Table 1). The SWC values

for the HG site were similar, however consistently slightly lower than those for the LG site. The highest values of SWC were in February–April 2018 (30 %) in the period with the highest precipitation, and the lowest in June–November 2019 (12 %), in the driest year over the measurement period.

The wind roses showed that the prevailing wind directions at the studied sites were NNW, N, NNE and SSE with the wind
speed mainly between 1–6 m s$^{-1}$ (Fig. 1). The footprint indicates that around 90% of the carbon fluxes measured by the EC

method were within the 70 m of the flux tower.

### 3.2 Diurnal patterns of carbon fluxes

The mean diurnal patterns of carbon fluxes are shown in Fig. 3 for each year and lumped together for the months of the dry
(June–December) and growing (January–May) seasons.

**Figure 3.** Mean diurnal carbon fluxes in the dry (a–d) and growing (e–h) seasons for years I–VI (top to bottom) in the (blue) lenient grazing
(LG) and (red) heavy grazing (HG) sites. Negative values indicate net carbon uptake, while positive values indicate net carbon release.
Shading area indicates ± 1σ.





During the dry seasons (June–December), the ecosystems were inactive (Fig. 3a, b, c, d). The carbon fluxes values were close to zero for the most part as net carbon uptake was constrained due to water unavailability and vegetation dormancy. Mean diurnal carbon exchange rates in the dry seasons were the lowest in the year I and the highest in the year II (Table 2).

**Table 2.** Summary of mean diurnal carbon fluxes in mg C m$^{-2}$ h$^{-1}$ of the dry (June–December) and growing (January–May) seasons for years I–IV. Years I–IV defined as hydrological year (November–October).

| | | $F_C$ | | | | | |
| | | min | | mean | | max | |
| | | LG | HG | LG | HG | LG | HG |
|---|---|---|---|---|---|---|---|
| Year I | growing season | −51.83 | −85.13 | 11.8 | 1.59 | 58.34 | 55.05 |
| | dry season | −11.14 | −13.31 | 4.0 | 4.13 | 12.94 | 14.03 |
| Year II | growing season | −135.92 | −165.43 | −11.09 | −25.01 | 68.62 | 62.33 |
| | dry season | 5.49 | 2.78 | 12.63 | 11.55 | 16.82 | 17.01 |
| Year III | growing season | −215.69 | −219.65 | −33.80 | −31.33 | 70.69 | 77.29 |
| | dry season | −11.34 | −27.89 | 6.2 | 4.19 | 17.85 | 24.23 |
| Year IV | growing season | −81.18 | −81.48 | 8.91 | 3.29 | 63.86 | 53.70 |
| | dry season | 7.35 | 5.97 | 10.13 | 9.05 | 13.01 | 12.59 |

During the growing seasons, the diurnal cycles of carbon fluxes for both sites demonstrate a classical behaviour with the carbon uptake predominating during the daytime and only respiration occurring during the night time (Fig. 3e, f, g, h).  Mean diurnal carbon fluxes showed a net carbon uptake in the growing seasons of years II and III with average values of –11.09 mg C m–2 h–1 and –25.01 mg C m–2 h–1 (year II), –33.80 mg C m–2 h–1 and –31.33 mg C m–2 h–1 (year III) for the LG and HG sites, respectively. Maximum daytime mean carbon release and uptake were both observed in the year III (Table 2).

The ecosystems turned from a carbon source to a carbon sink mostly from 6:00 to 18:00 LT, with the highest carbon sequestration values near midday (Fig. 4).  Midday uptake rates ranged from 48.15 mg C m$^{-2}$ h$^{-1}$ (year I) to 206.04 mg C m$^{-2}$ h$^{-1}$ (year III) for LG site and from 80.67 mg C m$^{-2}$ h$^{-1}$ (year I) to 218.98 mg C m$^{-2}$ h$^{-1}$ (year III) for the HG site.

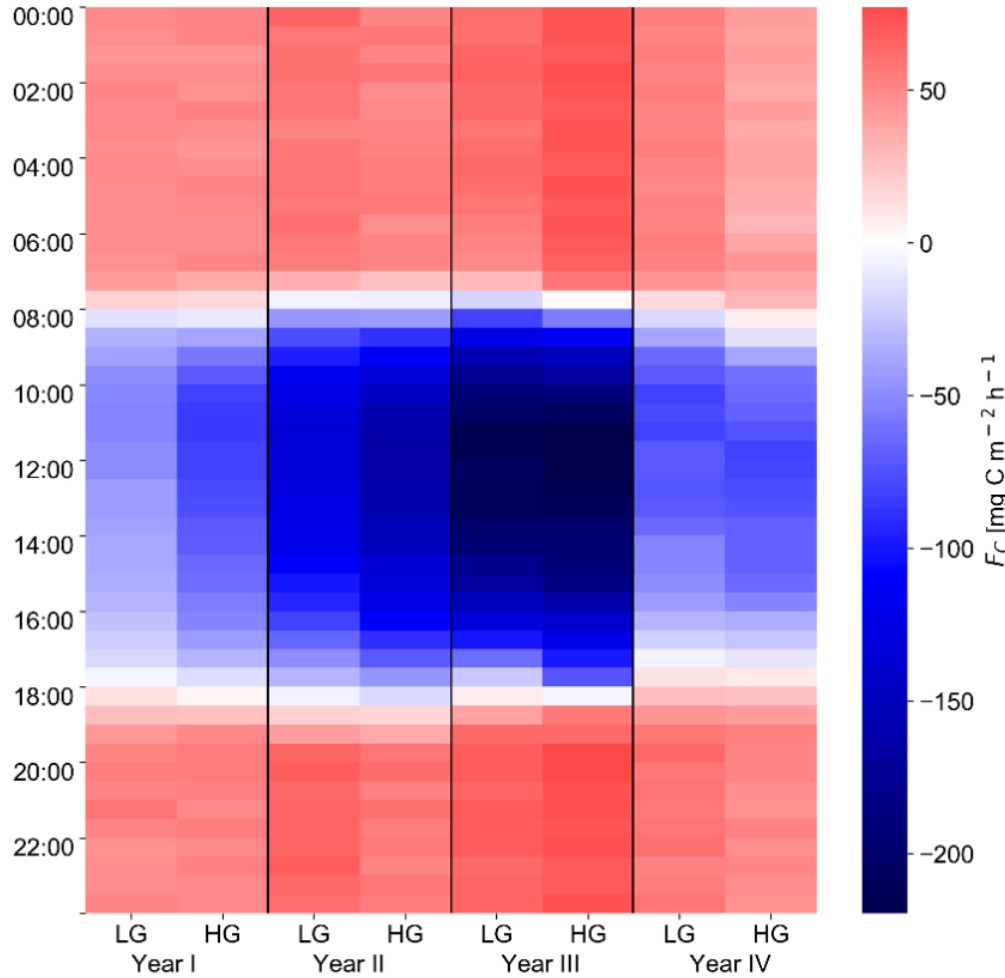

**Figure 4.** Mean diurnal carbon fluxes fingerprint in the growing seasons (January–May) in the lenient grazing (LG) and heavy grazing (HG)
sites. Blue colour represents net carbon uptake, while red indicates net carbon release.

### 3.3 Seasonal and annual NEE, GPP and $R_{eco}$ variations

The NEE, $R_{eco}$ and GPP at the studied sites varied seasonally with the peak values in the middle of the growing season
(February–March) (Fig. 5, 6). Their seasonal patterns followed the changes in precipitation and SWC (Fig. 2d). The growing
season lasted approximately 54 and 66 days (year I), 96 and 98 days (year II), 154 and 148 days (year III), and 95 and 96 days
(year IV) for the LG and HG sites, correspondingly (NEE < 0 on a daily basis). Throughout the measurement period, the
studied ecosystems acted as a net carbon sink of 440 and 511 days (16 months) for the LG and HG sites, respectively.





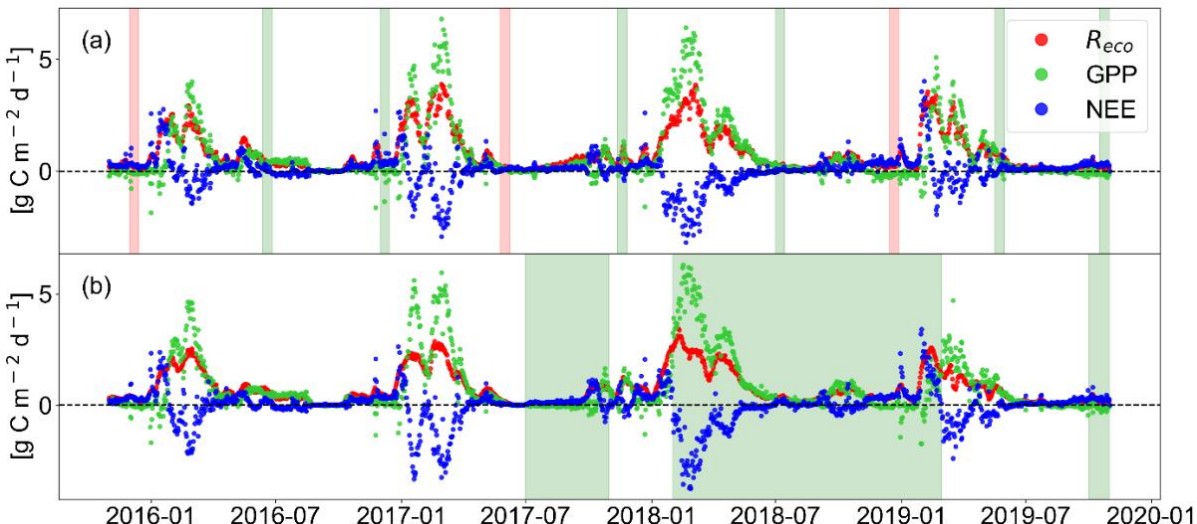

**Figure 5.** Daily cumulative measured net ecosystem exchange (NEE) and partitioned component fluxes (i.e. gross primary production (GPP), ecosystem respiration ($R_{eco}$) across different grazing intensities for (a) lenient grazing (LG) and (b) heavy grazing (HG) sites. The green patterns represent recorded livestock period and the red patterns represent estimated livestock period.

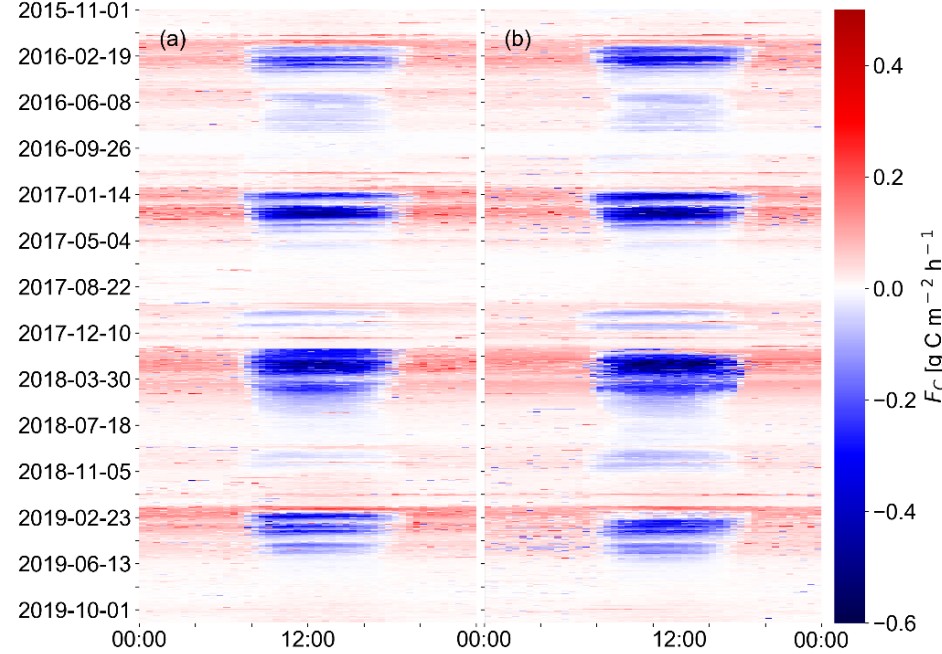

**Figure 6.** Temporal dynamics of the hourly carbon fluxes for the entire measurement period in the (a) lenient grazing (LG) and (b) heavy grazing (HG) sites.

The dry periods were mainly characterized by inactive ecosystems with low GPP, $R_{eco}$ and NEE. Mean values of carbon fluxes throughout the measurement period for the growing and dry seasons were –0.07 g C m$^{-2}$ d$^{-1}$ and 0.11 g C m$^{-2}$ d$^{-1}$ (LG), –0.15 g C m$^{-2}$ d$^{-1}$ and 0.10 g C m$^{-2}$ d$^{-1}$ (HG).



**Table 3.** Daily cumulative net ecosystem exchange (NEE), ecosystem respiration (R$_{eco}$) and gross primary production (GPP) in g C m$^{-2}$ d$^{-1}$. Years I–IV defined as hydrological year (November–October).

|  |  | NEE | | Reco | GPP |
|---|---|---|---|---|---|
|  |  | min | max |  |  |
| Year I | LG | −1.44 | 2.76 | 2.93 | 3.99 |
|  | HG | −2.17 | 2.39 | 2.54 | 4.64 |
| Year II | LG | −2.91 | 2.69 | 3.88 | 6.79 |
|  | HG | −3.31 | 2.64 | 2.88 | 5.96 |
| Year III | LG | −3.18 | 2.60 | 3.85 | 6.40 |
|  | HG | −3.74 | 2.06 | 3.44 | 6.31 |
| Year IV | LG | −1.93 | 4.02 | 3.54 | 5.09 |
|  | HG | −2.39 | 3.41 | 2.58 | 4.71 |

305

At the LG site, the magnitude of the daily NEE during the measurement period varied from –3.18 g C m$^{-2}$ d$^{-1}$ to 4.02 g C m$^{-2}$ d$^{-1}$. At the HG site, it ranged from –3.74 g C m$^{-2}$ d$^{-1}$ to 3.41 g C m$^{-2}$ d$^{-1}$ (Table 3).

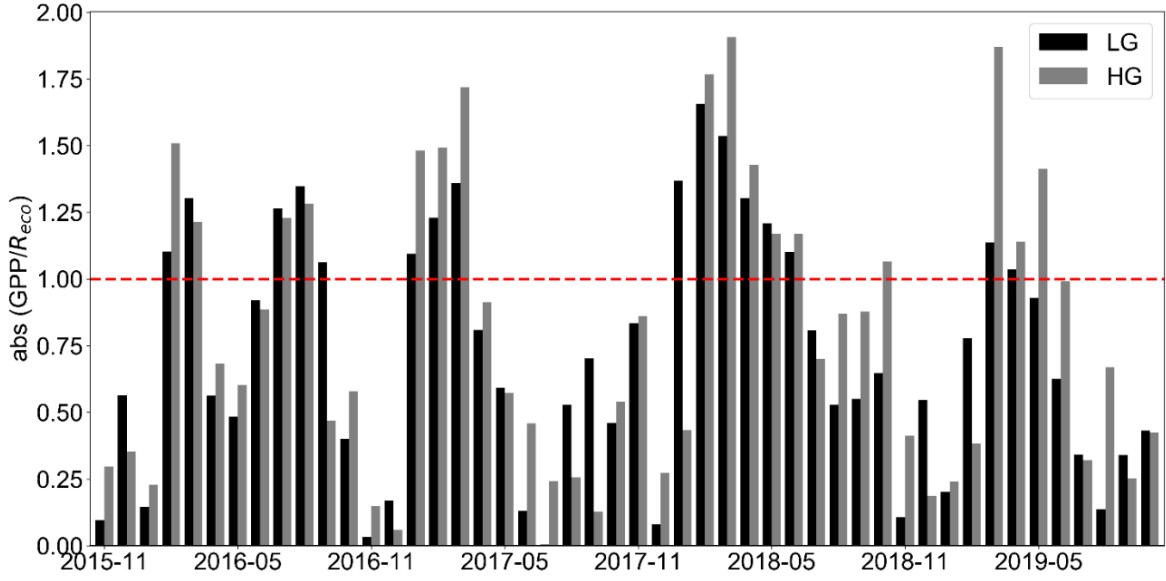

**Figure 7.** Absolute monthly ratios of gross primary production (GPP) versus ecosystem respiration (R$_{eco}$) in the (black) lenient grazing (LG) and (grey) heavy grazing (HG) sites. GPP/R$_{eco}$ > 1 ratios revealing a net carbon uptake.

At both sites, R$_{eco}$ was always highest during the growing season and decreased in the dry season with the lowest R$_{eco}$ in June–October. The highest daily sums of R$_{eco}$ were in the year III, whereas the lowest daily sums were in the year I for both sites (Table 3). Similarly, the GPP was highest during the growing season. During the dry seasons, the maximum GPP ranged from 0.39 g C m$^{-2}$ d$^{-1}$ to 1.43 g C m$^{-2}$ d$^{-1}$ for the LG site and from 0.58 g C m$^{-2}$ d$^{-1}$ to 1.65 g C m$^{-2}$ d$^{-1}$ for the HG site. Both R$_{eco}$





and GPP were higher at the LG site as compared to the HG site (Fig.5 a, b) with the absolute values of the GPP/$R_{eco}$ ratios revealing a net carbon uptake (GPP/$R_{eco}$ > 1) during most of the growing season (Fig. 7).

### 3.4 Carbon balance

Carbon budgets were estimated on the monthly, seasonal and annual scales to demonstrate the carbon source/sink strength of the studied sites. During the dry seasons, the LG site released on average 6.05 g C m$^{-2}$ month$^{-1}$, while the HG site released 5.30 g C m$^{-2}$ month$^{-1}$ (Fig. 8, Table 4). The highest carbon release for both sites was measured in December 2016 (year II). Dry season carbon sequestration on a monthly basis was observed only during July–August 2016 (cumulative precipitation 35 mm) (Table 4). Although only slight differences were found between the years I–IV in the dry seasons, significant fluctuations were observed during the growing seasons. The carbon sequestration periods varied throughout the measurement period in length (2 to 6 months) and in the strength of carbon sequestration. The monthly cumulative NEE showed the highest carbon sequestration rates in February 2018 for the LG site and in March 2018 for the HG site, and the highest carbon release in January 2016 for the LG site and in February 2019 for the HG site (Table 4).

**Table 4.** Monthly cumulative net ecosystem exchange (NEE) in g C m$^{-2}$. Years I–IV defined as hydrological year (November–October).

| | | Nov | Dec | Jan | Feb | Mar | Apr | May | Jun | Jul | Aug | Sep | Oct |
|---|---|---|---|---|---|---|---|---|---|---|---|---|---|
| Year I | LG | 8.54 ± 4.51 | 10.23 ± 4.49 | 41.18 ± 4.35 | −5.66 ± 5.45 | −13.18 ± 5.10 | 6.62 ± 4.64 | 14.17 ± 7.64 | 1.16 ± 10.29 | −2.64 ± 6.52 | −1.79 ± 6.98 | −0.04 ± 9.81 | 5.09 ± 6.95 |
| | HG | 6.93 ± 6.13 | 11.92 ± 4.53 | 31.84 ± 5.38 | −29.33 ± 5.94 | −10.99 ± 5.26 | 5.82 ± 4.62 | 8.47 ± 7.67 | 2.15 ± 11.44 | −2.53 ± 6.57 | −1.83 ± 6.26 | 0.51 ± 6.26 | 4.05 ± 5.77 |
| Year II | LG | 13.79 ± 4.25 | 23.68 ± 4.18 | −6.82 ± 4.96 | −17.01 ± 4.80 | −25.37 ± 6.51 | 3.16 ± 5.08 | 5.86 ± 4.87 | 3.40 ± 6.55 | 3.70 ± 6.29 | 3.88 ± 5.56 | 4.61 ± 4.54 | 11.82 ± 4.26 |
| | HG | 12.35 ± 4.24 | 20.50 ± 4.03 | −31.12 ± 5.66 | −29.23 ± 6.31 | −38.27 ± 7.01 | 1.56 ± 5.75 | 6.41 ± 4.92 | 0.69 ± 6.34 | 3.18 ± 6.66 | 4.19 ± 5.08 | 6.76 ± 4.97 | 11.64 ± 4.81 |
| Year III | LG | 2.52 ± 5.42 | 13.94 ± 4.24 | −15.08 ± 13.52 | −50.37 ± 6.00 | −35.66 ± 6.35 | −15.31 ± 7.50 | −6.08 ± 7.93 | −1.10 ± 5.53 | 1.37 ± 4.66 | 2.11 ± 5.41 | 6.18 ± 4.21 | 6.87 ± 4.20 |
| | HG | 3.24 ± 6.40 | 16.26 ± 6.93 | 25.46 ± 6.47 | −53.79 ± 20.75 | −54.27 ± 21.67 | −25.64 ± 21.42 | −5.28 ± 10.99 | −2.04 ± 6.46 | 2.73 ± 5.88 | 0.81 ± 5.33 | 1.93 ± 4.51 | −1.42 ± 4.30 |
| Year IV | LG | 9.69 ± 4.84 | 15.38 ± 4.99 | 22.07 ± 4.77 | 17.75 ± 6.66 | −8.23 ± 7.37 | −1.01 ± 5.51 | 1.72 ± 5.89 | 3.03 ± 4.60 | 3.51 ± 3.82 | 3.19 ± 5.96 | 7.35 ± 4.84 | 9.88 ± 5.22 |
| | HG | 7.01 ± 5.73 | 15.80 ± 4.10 | 22.32 ± 3.83 | 36.30 ± 9.30 | −34.57 ± 11.94 | −3.93 ± 5.71 | −8.20 ± 5.96 | 0.06 ± 4.22 | 3.67 ± 3.24 | 2.64 ± 5.33 | 7.63 ± 4.52 | 9.65 ± 4.53 |





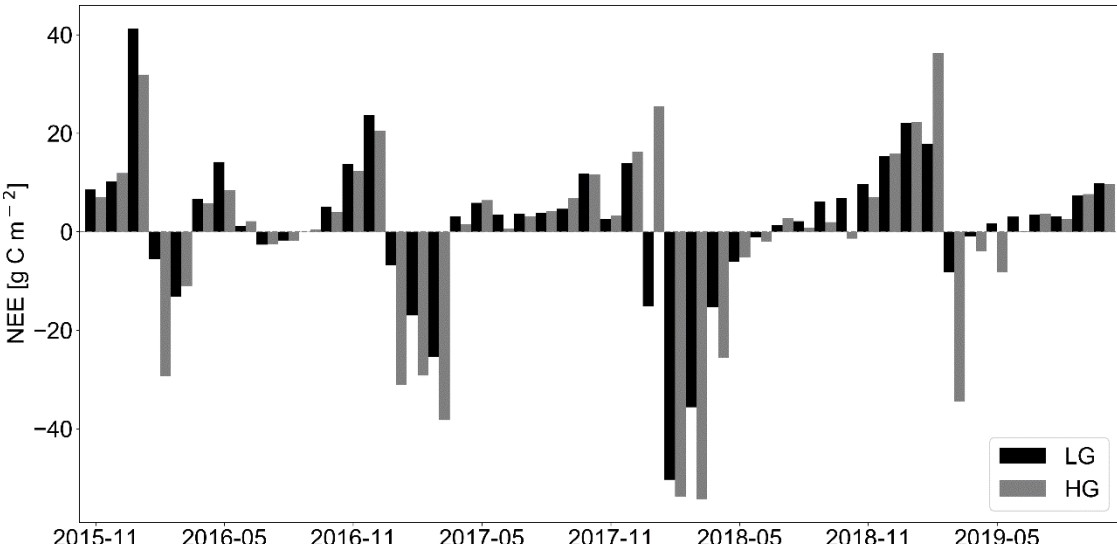

**Figure 8.** Monthly cumulative net ecosystem exchange (NEE) for the entire measurement period in the (black) lenient grazing (LG) and (grey) heavy grazing (HG) sites.

The seasonal NEE for the dry season of year I showed the lowest carbon release rates (Table 5). Similar NEE was observed during the dry season of year III. Meanwhile, dry season of year II demonstrated the highest carbon release with year IV demonstrating only slightly lower rates of carbon release (Table 5). During the growing seasons of years I and IV, carbon release was observed on a seasonal basis. In comparison, years II and III showed enhanced carbon sequestration with the highest carbon uptake in the year III for both sites.

**Table 5.** Seasonal net ecosystem exchange (NEE) and precipitation (P) in the growing (January–May) and dry (June–December) seasons. Years I–IV defined as hydrological year (November–October).

| | | NEE (g C m$^{-2}$) | | P (mm) | |
|---|---|---|---|---|---|
| | | LG | HG | LG | HG |
| | growing season | 43.13 ±20.65 | 5.81 ±21.45 | 318.5 | 316.1 |
| Year I | dry season | 20.54 ±37.76 | 21.22 ±34.75 | 47.7 | 46.2 |
| | growing season | −40.17 ±18.85 | −90.65 ±21.63 | 296.6 | 306.8 |
| Year II | dry season | 64.87 ±28.24 | 59.32 ±28.89 | 68.3 | 67.4 |
| | growing season | −122.51 ±30.04 | −113.54 ±43.34 | 401.9 | 377.8 |
| Year III | dry season | 31.89 ±25.98 | 21.51 ±28.89 | 85.6 | 80.4 |
| | growing season | 32.31 ±21.26 | 13.42 ±25.45 | 286.3 | 257.7 |
| Year IV | dry season | 52.03 ±25.99 | 46.47 ±30.89 | 8.2 | 9 |


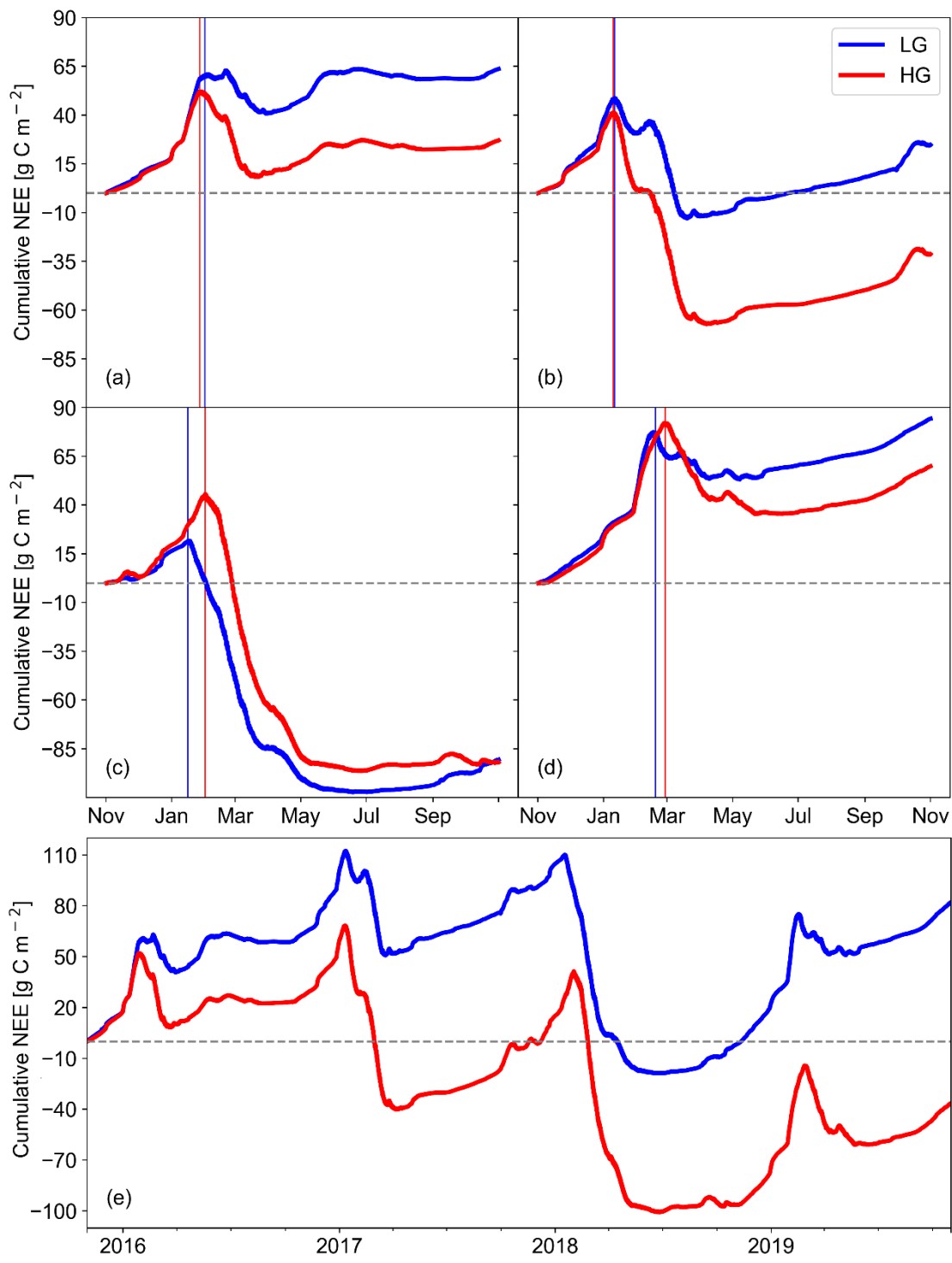

**Figure 9.** Annual cumulative net ecosystem exchange (NEE) (a–d) for the years I–IV with vertical lines indicating the highest respiration point throughout the year when ecosystem turning into a carbon sink and (e) four years cumulative NEE in the (blue) lenient grazing (LG) and (red) heavy grazing (HG) sites.





Annually integrated cumulative NEE varied from carbon sink to source over the four years measurement period (Fig. 9). Only year III of the four years had negative NEE (i.e. were net carbon sinks at the annual timescale) for the LG site and years II–III for the HG site. In the year I, the total cumulative NEE was higher for the LG site ($63.68 \pm 55.59$ g C m$^{-2}$ yr$^{-1}$) compared to the HG site ($27.03 \pm 53.24$ g C m$^{-2}$ yr$^{-1)}$. In the year II, however, a sequestration of $31.33 \pm 47.44$ g C m$^{-2}$ yr$^{-1}$ was measured for the HG site compared to a release of $24.69 \pm 44.27$ g C m$^{-2}$ yr$^{-1}$ at the LG site. In the year III, both sites acted as carbon

sinks with sequestration rates of $90.61 \pm 51.39$ g C m$^{-2}$ yr$^{-1}$ (LG) and $92.02 \pm 66.73$ g C m$^{-2}$ yr$^{-1}$ (HG). In year IV, impacted by drought, carbon uptake was limited by water availability even during the peak of the wet season in summer. It resulted in carbon release rates of $84.33 \pm 43.76$ g C m$^{-2}$ yr$^{-1}$ (LG) and $59.89 \pm 46.11$ g C m$^{-2}$ yr$^{-1}$ (HG). After the four years measurement period, the total cumulative NEE indicated a carbon release of $82.11 \pm 48.75$ g C m$^{-2}$ for the LG site while the HG site was a slight carbon sink with a sequestration rate of $36.43 \pm 53.38$ g C m$^{-2}$.

**3.5 Correlation between NDVI and GPP**

We investigated the relationship between each NDVI (each 16 days) and the corresponding daily sums of GPP (Fig. 10). High leaf area leads to a higher light absorption capacity, and hence result in a higher GPP when water availability is sufficient. The NDVI values range from zero (no vegetation) to 1 (dense vegetation).

**Table 6.** Statistical summary of normalised difference vegetation index (NDVI) in the growing (January–May) and dry (June–December) seasons. Years I–IV defined as hydrological year (November–October).

| | | NDVI | | | | | |
|---|---|---|---|---|---|---|---|
| | | LG | | | HG | | |
| | | min | mean | max | min | mean | max |
| Year I | growing season | 0.20 | 0.31 | 0.41 | 0.21 | 0.33 | 0.52 |
| | dry season | 0.19 | 0.23 | 0.30 | 0.19 | 0.25 | 0.29 |
| Year II | growing season | 0.25 | 0.38 | 0.63 | 0.24 | 0.40 | 0.72 |
| | dry season | 0.19 | 0.21 | 0.25 | 0.19 | 0.21 | 0.24 |
| Year III | growing season | 0.19 | 0.43 | 0.61 | 0.19 | 0.44 | 0.68 |
| | dry season | 0.19 | 0.23 | 0.29 | 0.19 | 0.22 | 0.26 |
| Year IV | growing season | 0.19 | 0.35 | 0.48 | 0.20 | 0.36 | 0.53 |
| | dry season | 0.19 | 0.21 | 0.24 | 0.19 | 0.21 | 0.23 |

Overall mean NDVI in the dry seasons was 0.22 for both sites (Fig. 2e, Table 6) with the lowest NDVI values during spring (September–November). The mean values in the growing seasons were 0.37 and 0.38 for the LG and HG sites, respectively.

The maximum NDVI values were measured in mid-February in the years II and III (Table 6). Mean annual NDVI values were



0.27 and 0.28 in the year I, and 0.29 and 0.30 in the year II for LG and HG sites, respectively. In the years III and IV, the mean NDVI values were 0.32 and 0.27 for both sites.

Analysis of the seasonal trends of NDVI and derived GPP demonstrated strong linear correlations (p-value <0.0001) between them for both studied sites (Fig. 10). The coefficient of determination ranged from 0.65 to 0.93 for the LG site and from 0.71

to 0.96 for the HG site.

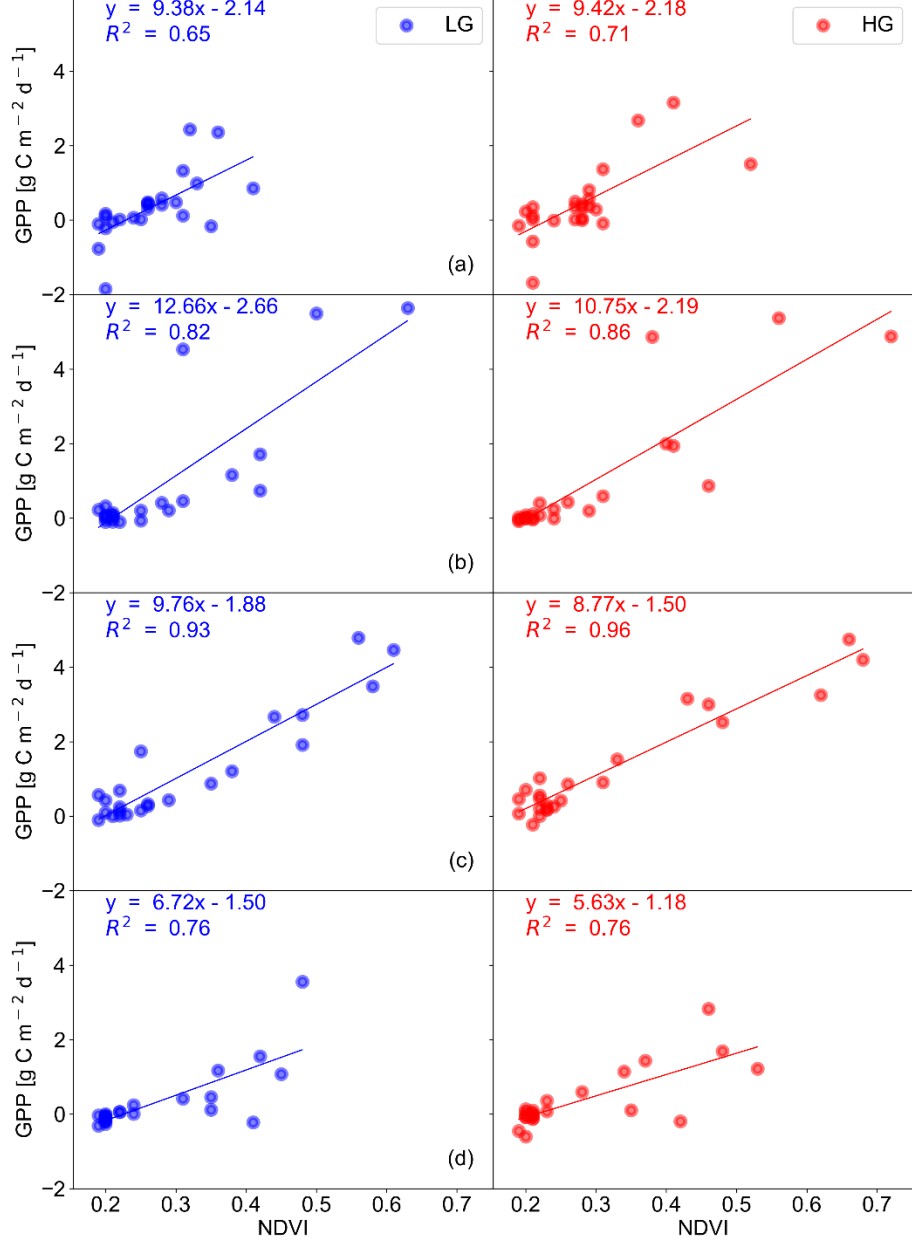

**Figure 10.** Correlation between normalised difference vegetation index (NDVI) and gross primary production (GPP) for the years I–IV (a–d) in the (blue) lenient grazing (LG) and (red) heavy grazing (HG) sites.



## 4 Discussion

This study provides new understanding of the impact of current and past grazing intensities on $CO_2$ fluxes and their daily, seasonal, and annual dynamics in semi-arid South African ecosystems. We found that the two investigated grazing regimes under similar climate, soil conditions and topography have highly influenced plant species composition and vegetation cover leading to implications for their role as potential grazing areas and/or efficient $CO_2$ sinks in the future.

### 4.1 Effect of livestock grazing on carbon sink strength

**4.1.1 Impacts of overgrazing via altered vegetation cover**

After four years of measurements, our results indicate that the HG ecosystem was unexpectedly a slight carbon sink, while the LG ecosystem was a considerable carbon source, thereby disproving our hypothesis of the LG site being a stronger sink for atmospheric $CO_2$. The differences in carbon fluxes between the studied sites could be explained by different vegetation cover (species distribution as well as aboveground biomass) due to overgrazing in the past. During the resting period, the HG site
showed consistently slightly higher mean NDVI and had slightly higher NDVI peaks compared to the LG site. Increases in canopy cover after a long resting period were reported by several authors (Chen et al., 2014; Lin et al., 2011; Mofidi et al., 2013). Yan et al. (2014) reported that long-term heavy livestock grazing resulted in significant decrease of soil C and N storage, which may significantly alter the ecosystem causing decrease in the carbon sequestration rates. The HG system is dominated by unpalatable grasses, while nearly all palatable shrubs have been extirpated by overgrazing. Zhou et al. (2012) showed that
the fine root biomass of perennial grasses was more abundant in the upper soil layer compared to shrubs. It implies that grasses are stronger competitors for water, especially in water-limited ecosystems with pulsed precipitation. This coincides with our finding that in the spring (September–November), the studied sites were net sources of carbon for all measurement years, with the exception of the HG site during October 2018 with 15 mm of precipitation (Table 4, Fig. 8). The difference was likely due to the vegetation cover at the HG site, being dominated by grasses, and thus able to respond more quickly to rain events due
to shallow-root systems that use discontinuous and erratic water sources in the upper soil layers compared with the shrubs deep-root systems that use water in deeper soil layers (Canadell et al., 1996; Hipondoka et al., 2003; Zhou et al., 2012). *Aristida diffusa* is a drought tolerant dominant grass at the HG site that managed to survive the intense summer drought, although net mortality of other grass species was observed (du Toit and O'Connor, 2020). Furthermore, Zhou et al. (2012) found that total soil organic carbon storage was higher for perennial grasses (higher soil organic carbon inputs through primary production and
slower return of carbon through decomposition) than for shrubs.

The differences in carbon budgets between the studied sites indicate that a long resting period (2007–2017) along with transition to unpalatable drought resistant grass species at the previously overgrazed site improved the carbon sink potential (HG site had higher carbon sequestration rates compared to the LG site) due to an increase in the canopy cover.  At the same time, the site is unfavourable from the perspective of the current livestock grazing system, and restoring its original species
composition would return its value as pasture. Leu et al. (2014) showed that heavily grazed shrubland (northern Negev, Israel)





can be recovered within <16 years by implementing strict conservation management. Seymour et al. (2010) reported that 20 years of recovery period in the Karoo degraded ecosystems restored grazing potential, while not returning all palatable species.

### 4.1.2 Impacts of lenient temporary vs. continuous heavy livestock grazing

At the HG site, a slight decrease in the carbon uptake peaks was linked to an increase of continuous grazing time (Figs. 3, 5).
Figure 3 shows that during the resting period (I and II years), the HG site had a significantly higher mean diurnal NEE compared to the LG site with rotational grazing. However, during the years III and IV, the mean diurnal NEE patterns were similar for both sites. This could be explained by continuous livestock grazing, which began in July 2017 at the HG site. The impacts of livestock grazing on the ecosystems  were associated in previous studies with canopy cover, the response of the dominant plant species to grazing, different grazing regimes, and site history (Chen et al., 2014; Gan et al., 2012; K'Otuto et
al., 2013; Kairis et al., 2015; Lecain et al., 2000; Räsänen et al., 2017; Tagesson et al., 2015; Talore et al., 2016; Wagle et al., 2019; Yan et al., 2017). In a study conducted in Chinese and Mongolian steppes, Na et al. (2018) showed that continuous grazing leads to reduction in aboveground biomass (AGB).  Ma et al. (2019) found a strong correlation between grazing intensity and vegetation indexes (AGB and NDVI). At high grazing intensity, canopy cover and vegetation indexes showed a decreasing trend (Amiri et al., 2008; Belgacem et al., 2013; Ping et al., 2018; Sun et al., 2011) resulting in a higher percentage
of the soil surface exposed to radiation, followed by increased evaporation and decrease in SWC (Chen et al., 2013; Villegas et al., 2014; Yan et al., 2016, 2017). In our study, analysis of the NDVI trends supported the interpretation of the impacts of continuous grazing: mean NDVI values decreased in the HG site due to persistent livestock grazing and resulted in the same mean values for both sites in the years III and IV (Sect. 3.5, Fig. 2e).

Our results showed a strong positive correlation between the NDVI and the daily sums of GPP for both studied sites (Fig. 10).
Previous studies also reported a strong linear correlation between the GPP and NDVI in the grassland, shrubland and savanna ecosystems (Huang et al., 2019; Yan et al., 2019). The seasonal GPP peaks followed the course of the NDVI variations. The HG site had slightly higher coefficients of determination in the Years I–III than the LG site. The natural state of the HG site was disturbed by heavy livestock grazing followed by a change in species composition to unpalatable grass species, which made the site robust concerning the drought conditions. In the Year IV, both sites had similar correlations. It shows that the
long continuous grazing at the HG site has a negative impact on the vegetation cover, which also affects carbon exchange.

### 4.2 Water availability as the main driver of inter-annual variability of carbon fluxes

While we could attribute differences in $CO_2$ exchange between sites to grazing intensity, sums and their variability on an annual basis were still highly dominated by the amount of available water, as reported in previous studies conducted in similar systems (Merbold et al., 2009 and references therein). During the growing seasons, carbon fluxes were negative in the daytime
(correlated with light intensity throughout the day) and positive at nighttime (changed to carbon release after the sunset). At the beginning of the growing season, during the onset of summer precipitation events, an immediate response of $R_{eco}$ to precipitation and plant germination was observed. First precipitation event also caused an increase in GPP to some extent, but





the NEE was comparatively small during the plant germination phase due to high respiration. However, during the peak of the growing season, GPP exceeded $R_{eco}$ and the ecosystems turned into a carbon sink. This is consistent with other findings in the

semi-arid ecosystems (Kutsch et al., 2008; Merbold et al., 2009; Tagesson et al., 2015, 2016; Williams and Albertson, 2005). During the dry seasons, the ecosystems were not physiologically active due to the limited water availability. These findings are in line with other studies conducted in African ecosystems (Archibald et al., 2009; Brümmer et al., 2008; Räsänen et al., 2017; Tagesson et al., 2015; Veenendaal et al., 2004). Mean carbon fluxes showed the highest carbon uptake and release rates in the year III, followed by the year II. It is in line with the temporal distribution of precipitation and NDVI indexes. Based on

NDVI data, year II stands for the highest NDVI peak, whereas year III represents the wettest year with the longest NDVI peaks (the longest growing season), years I and IV represent the lowest NDVI peaks due to the distribution (year I) and the deficit of precipitation (year IV).

At the annual scale, the carbon sink strength in the studied sites varied annually depending on the climatic conditions. Years I and II had similar precipitation rates (373 mm and 370 mm) (Fig. 2d, Table 1). The LG site acted as a carbon source in both

periods but in the year II with lower strength compared to the year I, whereas the HG site was a carbon source in the year I and carbon sink in the year II (Fig. 9). The differences in the carbon sequestration rates between years I and II were determined by different precipitation distribution which resulted into the short growing season in the year I. During the transition period (November–December) sites received only 10 mm of precipitation in the year I compared to the year II (47 mm). During the peak of the growing seasons (January–March), both sites received 230 mm and 280 mm of precipitation for the years I and II,

respectively. At the end of the growing season (April–May), both sites had higher amount of precipitation in the year I (88 mm) than in the year II (23 mm). During the dry months (June–October) of year I, both sites received 47 mm of precipitation, which explains carbon sink months in the dry period (July–August), while during the dry season of the year II, sites received only 20 mm. Fig.2d shows that the same amount falls within two rainy periods in the year II causing a longer and delayed soil moisture decay leading to a longer growing season than in year I, where more equally distributed rain led to a faster dry up. In

the year III, both sites acted as considerable carbon sinks (Fig. 9) due to the increased precipitation and, as a result, increased length of the growing season (Table 1). In the year IV, the studied ecosystems acted as considerable carbon sources with the highest $CO_2$ release due to the precipitation deficit compared to the long-term mean annual precipitation (approximately 25 %). Our results agree with other, previously mentioned studies, which observed similar seasonal dynamics of NEE in their respective regions (Brümmer et al., 2008; Jongen et al., 2011; Quansah et al., 2015; Tagesson et al., 2015, 2016; Veenendaal

et al., 2004; Xu and Baldocchi, 2004).

Similar annual NEE of –98 g C m$^{-2}$ yr$^{-1}$ to 21 g C m$^{-2}$ yr$^{-1}$ were observed by Scott et al. (2010) in the Kendall grassland, USA with mean annual precipitation of 345 mm (63 % in the summer months). Räsänen et al. (2016) observed annual carbon budgets of – 85 g C m$^{-2}$ yr$^{-1}$, 67 g C m$^{-2}$ yr$^{-1}$ and 139 g C m$^{-2}$ yr$^{-1}$ (2011–2013) in the Welgegund atmospheric measurement station grassland ecosystem, South Africa (540 mm). The annual NEE ranges at the South African Kruger National Park were from –

138 g C m$^{-2}$ yr$^{-1}$ to 155 g C m$^{-2}$ yr$^{-1}$ with a mean annual precipitation of 550 mm (Archibald et al., 2009). Niu et al. (2020)





reported a carbon source ecosystem with carbon release ranging from 34.99 C m$^{-2}$ yr$^{-1}$ to 63.05 g C m$^{-2}$ yr$^{-1}$ in a semi-arid sandy grassland ecosystem in northern China (360 mm).

On a broader context, Valentini et al. (2014) reported a small carbon sink of 0.61 ± 0.58 Pg C yr$^{-1}$ (−20.1 g C m$^{-2}$ y$^{-1}$) on a whole continent on an annual basis by averaging out all the estimates. This can be compared with measured here mean annual

NEE of 20.57 g C m$^{-2}$ y$^{-1}$ for LG site and −9.11 g C m$^{-2}$ y$^{-1}$ for HG site. However, complete and accurate estimation of the carbon budget for the African continent is not available due to large gaps in carbon fluxes data from in situ measurements for many African ecosystems. Thus, understanding the dynamics of carbon fluxes in different types of ecosystems is crucial to underpin these larger-scale estimations.

## 5 Conclusions

We analysed four years of measured $CO_2$ fluxes from 1 November 2015 to 31 October 2019 at two semi-arid Karoo (South Africa) ecosystems with different grazing intensities. To our knowledge, the current study provides the first long-term EC measurement of carbon fluxes, which are compared between semi-arid ecosystems with similar climatic conditions but different past and present grazing intensity in South Africa.  Unexpectedly, the site with continuous heavy grazing after a long resting period was more efficient as a $CO_2$ sink. This was linked to a higher abundance of unpalatable grasses, which may be

better able to compete for water. At the same time, the high ratio of unpalatable vs palatable species made this site less suitable for its current use as sheep pasture. This may signify that an optimally managed grazing site is not necessarily the most carbon efficient ecosystem. Slight decreases in the carbon uptake peaks were observed in response to resuming heavy grazing at the HG site (years III–IV) by modification of the SWC, soil properties and aboveground biomass.

 The interannual variability of carbon budgets was mainly driven by the annual amount and seasonal distribution of

precipitation. The annual cumulative NEE varied from –92.02 g C m$^{-2}$ yr$^{-1}$ (year III at the HG site) to 84.33 g C m$^{-2}$ yr$^{-1}$ (year IV at the LG site) demonstrating that semi-arid Karoo ecosystems under livestock grazing can act as a carbon source or carbon sink, depending on meteorological (precipitation) conditions. Our study contributes to a better understanding of livestock grazing effects on the ecosystem–atmosphere carbon exchange in the semi-arid Karoo ecosystems. It may have important implications on the design of sustainable grazing strategies in the region.

## Appendix A. Latent and sensible heat fluxes and energy balance closure


Half-hourly time-series of the latent and sensible heat fluxes are shown in Figure A1. During February-April (growing season) turbulent energy fluxes were dominated by latent heat flux, while during May–January they were dominated by sensible heat flux. During the growing season, the night-time peaks for the latent and sensible heat fluxes were 90 W m$^{-2}$ and 45 W m$^{-2}$, while daytime peaks reached 400 W m$^{-2}$ and 470 W m$^{-2}$ (Table A1). Peak latent and sensible heat flux values in the dry seasons

were 55 W m$^{-2}$ and 75 W m$^{-2}$ during night-time, and reached the maximum of 250 W m$^{-2}$ and 500 W m$^{-2}$ during daytime. The



four-year means of H, LE, G and Rn were 65 W m⁻², 24 W m⁻², 15 W m⁻², 213 W m⁻² for the LG site and 71 W m⁻², 24 W m⁻², 14 W m⁻², 250 W m⁻² for the HG site. Also, Rn was slightly higher in the dry seasons than in the growing seasons.

**Table A1. Summary of energy balance components: means of sensible heat (H), latent heat (LE), ground heat flux (G), net radiation**
**(Rn) in the growing (Jan–May) and dry (Jun–Dec) seasons. Years I–IV defined as hydrological year (Nov–Oct).**

|  |  | H (W m⁻²) | | LE (W m⁻²) | | G (W m⁻²) | | Rn (W m⁻²) | |
| --- | --- | --- | --- | --- | --- | --- | --- | --- | --- |
|  |  | LG | HG | LG | HG | LG | HG | LG | HG |
| Year I | growing season | 48 | 49 | 37 | 38 | −0.37 | 1.67 | 105 | 116 |
|  | dry season | 59 | 62 | 8 | 9 | 3.87 | 6.89 | 99 | 113 |
| Year II | growing season | 44 | 51 | 39 | 37 | 0.20 | −5.51 | 110 | 126 |
|  | dry season | 60 | 66 | 11 | 11 | 9.47 | 4.15 | 132 | 142 |
| Year III | growing season | 47 | 55 | 41 | 42 | 0.93 | 1.41 | 119 | 152 |
|  | dry season | 59 | 61 | 12 | 16 | 4.28 | 3.73 | 95 | 113 |
| Year IV | growing season | 52 | 52 | 31 | 32 | 0.90 | 1.21 | 105 | 117 |
|  | dry season | 66 | 68 | 7 | 8 | 4.32 | 6.17 | 87 | 95 |

The partitioning of the turbulent energy can be additionally analysed by examining the H/Rn and LE/Rn ratios. During the peak of the growing seasons, the H/Rn ratios were 0.22 and 0.27, whereas during the dry periods, those ratios were 0.70 and 0.68 for the LG and HG sites, respectively. The LE/Rn ratios during the growing and dry periods were 0.73 and 0.18 for the
LG site, and 0.65 and 0.18 for the HG site.

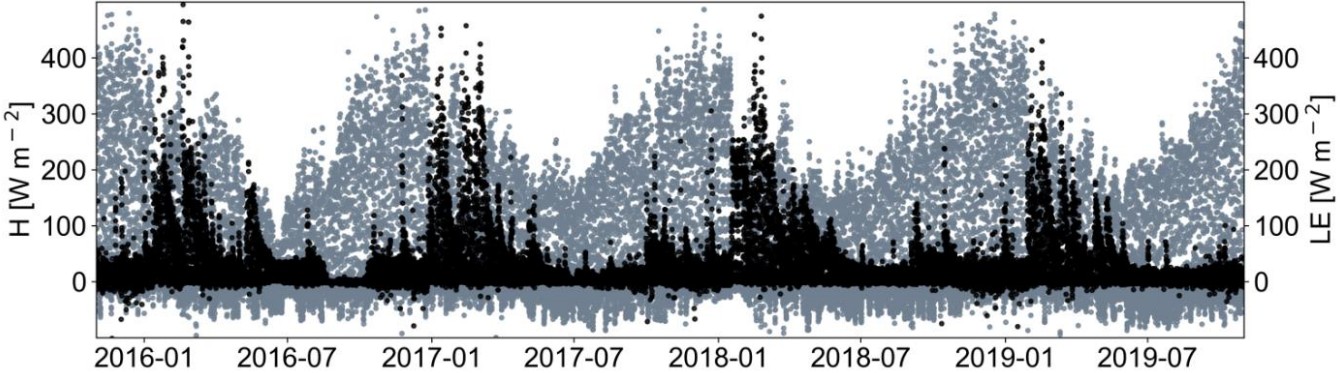

**Figure A1.** Half-hourly time-series of the sensible heat (H, grey dots) and latent heat (LE, black dots) fluxes.



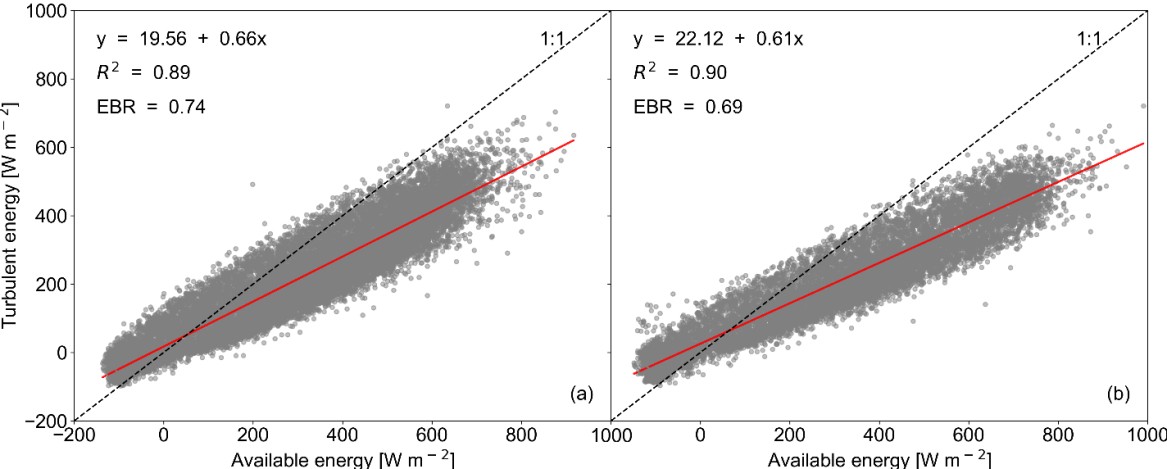

**Figure A2.** Energy balance closure (EBC) determined by turbulent energy fluxes (H + LE + SH + SLE) versus available energy (Rn – G – Sg): in the (a) lenient grazing (LG) and (b) heavy grazing (HG) sites.

The half-hourly data of G, Rn, H and LE were used to evaluate and analyse EBC for the entire study period based on the linear regression statistics between the turbulent and available energy (Fig. A2). The numbers of half-hourly data used for EBC evaluation were 19,698 and 13,955 for LG and HG sites, correspondingly. No seasonal trends were found. The regression fits between turbulent energy and available energy reveal a strong linear relation with coefficients of determination ($R^2$) of 0.89 and 0.90 for the LG and HG sites. The intercepts and slopes were 19.56 and 0.66 for the LG site, and 22.12 and 0.61 for the HG site, respectively. The EBRs for the entire study period were 0.74 for the LG site and 0.69 for the HG site. The residual of the surface energy balance during daytime ranged from –154 W m$^{-2}$ to 479 W m$^{-2}$ and from –169 W m$^{-2}$ to 513 W m$^{-2}$ with highest values between 10:00 and 12:00 for the LG and HG sites, respectively (Fig. A3). During the night-time the residual varied from –132 W m$^{-2}$ to 56 W m$^{-2}$ and from –134 W m$^{-2}$ to 91 W m$^{-2}$ for the LG and HG sites.

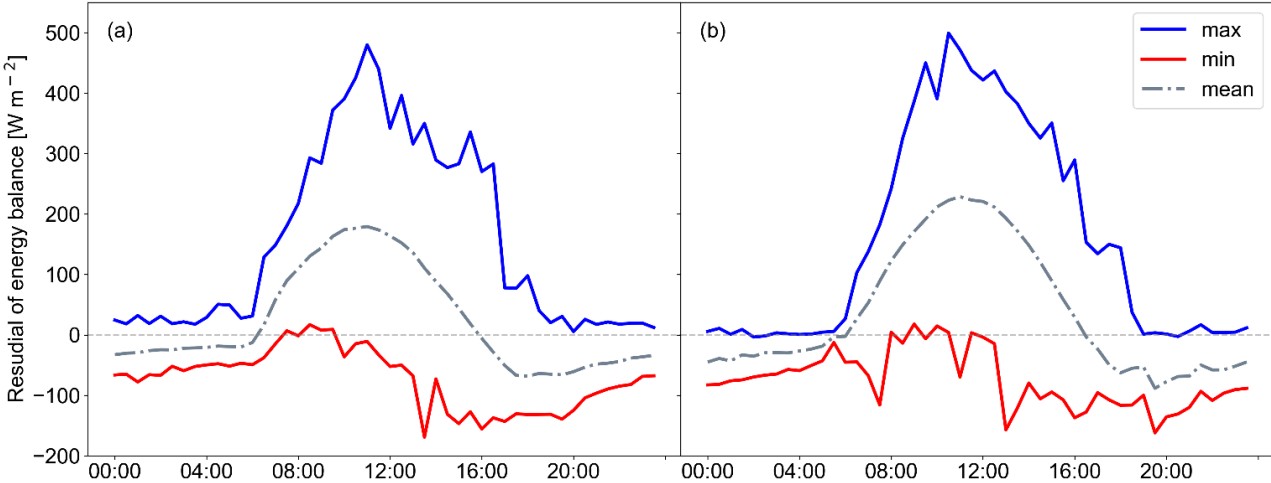



**Figure A3.** Maximum (blue), mean (grey) and minimum (red) diurnal variation of the residual of the energy balance in the (a) lenient grazing (LG) and (b) heavy grazing (HG) sites.

*Data availability.* Data used in this study can be requested from Oksana Rybchak (oksana.rybchak@thuenen.de). Furthermore,
the data will be uploaded to the Fluxnet database.

*Author contributions.* CB, GF and OR conceived the study. JP, KM, JKJ, JdT installed and managed the eddy covariance flux
towers, collected the raw data. JdT provided information about the studied sites (biodiversity, grazing management) and helped
to write a corresponding sub-section in the methodology part. CT provided the remote sensing-based vegetation MODIS
indices and wrote a corresponding sub-section in the methodology part. CB, GF, MB gave scientific advice to the overall data
analysis and interpretation. OR wrote the manuscript, processed raw data, conducted flux data analysis and interpretation. All
authors discussed and reviewed the manuscript.

*Competing interests.* The authors declare that they have no conflict of interest.

*Acknowledgements.* The authors acknowledge funding from the German Federal Ministry of Education and Research (BMBF),
framework programmes SPACES and SPACES II (Science Partnerships for the Assessment of Complex Earth System
Processes in Southern Africa), projects ARS AfricaE (grant number 01LL1303A) and EMSAfrica (grant number 01LL1801E).

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
