# Peer review of "Multi-year CO2 budgets in South African semi-arid Karoo ecosystems under different grazing intensities"

_Biogeosciences, 2020_

## Referee Comment (RC1) · Marc Aubinet (Referee) · 4 Jan 2021

Synthesis

The paper presents CO2 budgets of semi-arid grasslands/shrubs ecosystems in Karoo, South Africa with a special focus on grazing. Two sites, one considered as submitted to heavy grazing, the other to a lenient grazing, were compared. CO2 fluxes were measured using the eddy covariance technique.

Semi-arid ecosystems and, more generally, African ecosystems are not enough investigated and poorly covered by flux measurements. In addition, as measures are sought

to mitigate climate change, especially in the livestock sector, it is of primary importance to better understand and quantify the impact of cattle management on ecosystem carbon balance. I think thus that the topic is of great interest. In addition, it perfectly fits the scope of the journal.

The methodology used to measure and treat the eddy covariance data seems correct and well implemented. I have however one restriction concerning the HG site that I will detail below.

The general paper shape (quality of writing, figure quality, reference list) is good.

I thus think that the measurements realised here have the potential to make a good paper that would deserve publication.

However, I have a strong reservation concerning the result presentation and discussion. The results have not been sufficiently addressed, being most of the time limited to a chronological presentation of the fluxes. The analysis is too superficial, premature and based on an over-interpretation of the results, I will detail this below. I thus think that the paper needs a complete reshape and, in view of the importance of the needed changes, I suggest rejection of the paper and I encourage the authors to resubmit it after a complete redesign of their result presentation and analysis.

General comments

The two main theses in this paper are that (i) site to site differences between carbon budgets is mainly determined by the current and past management and that (ii) their interannual variability is linked to soil humidity.

The first point is announced from the beginning and repeated several times in the paper but, unfortunately, never really supported by the measurements:

First, the budget over four years yields to 82.11 and -36.43 gCm-2 (corresponding to annual budgets of 20.5 and -9.1 gCm-2yr-1) for LG and HG, respectively. In view of the uncertainties due to the measurement technique and of the interannual variability,

I doubt that these budgets are significantly different and, moreover, that each budget is significantly different from zero. The interpretation (which is central to the paper) that one site behaves as a source and the other as a sink is thus abusive. It is most probable that none of the site significantly depart from equilibrium.

Secondly, even if a significant site to site difference had to be found (maybe on the two first years), it could be attributed as well to present as to past management. Even if a possible impact of present management is evoked at some place, the reader is leaded to think that the past management is the most important factor. Indeed, in the material and method, a description of present management is missing and we have to wait for the result presentation (Figure 5) to realize that present management differed completely, not only between the two sites but also between the beginning and the end of measurement period in one site, the so-called "heavy grazed" site being not grazed at all during the two first years (!!). As a result the impact of presented grazing on the flux has been completely overlooked, which biases the conclusions and the discussion.

The analysis of the long term impact of overgrazing based on the differences between parcel species composition, with a greater abundance of unpalatable grasses in the HG site, is promising. Unfortunately, the analysis is only qualitative and does not rely on a quantitative inventory of species on each site.

Concerning the second point (response to humidity), it would be relevant to attribute the interannual variability to precipitation and soil moisture differences. This aspect is however insufficiently developed to my opinion and results presenting the responses of NEE, GPP and Reco to SWC should be deepened.

Finally, I have some reservations as regards the measurement methodology on the HG site for the grazed years: the authors announced that the site was divided in two paddocks and that the cattle were regularly moved from one paddock to another. I suppose thus that CO2 fluxes should differ if they come from the occupied or from the empty paddock due to cattle respiration but also to vegetation consumption. However,

no information about the paddock positioning with respect to the measurement tower or about cattle moving chronology was given; no flux discrimination between the empty and the occupied paddock was tempted. In these conditions, I suspect that the fluxes should be strongly affected by footprint variations. Discriminating fluxes between the paddocks, if possible, could provide information on the impact of present grazing on fluxes.

I recommend the authors reading papers by Jerôme et al., AEE 194, 7-16, 2014; Felber et al., BG 13, 2959-2969, 2016; Gourlez de la Motte et al., AFM 268, 156-168, 2019, and references herein to better address the difficult question of highlighting the grazing impact on carbon balance.

More specific comments

Abstract

L21: As already said above, in view of the uncertainties and of the interannual variability, I think that the difference between the two sites is not significant (and a source of 82 gC m-2 over 4 years is by no means "considerable" as the uncertainty on the annual sequestration is about 50 gC m-2yr-1). I found thus misleading to present one site as a source and the second as a sink. Both sites do probably not depart significantly from equilibrium. As this assertion is central to the paper, it questions all the discussion and conclusions.

Introduction

L78-80, Independently of the fact that I disagree with the sentence, an introduction should finish with a presentation of the objectives of the papers in the form of scientific questions, not with a result.

Sites description

A section describing the cattle management during the measurement period (stocking rate evolution, chronology, position of the paddocks) is missing. The fact that the socalled "heavy grazed" site is not grazed at all during the first two years should appear more clearly (and the name of the site should be changed as it is strongly misleading!).

What are the parcel sizes?

L100: The stocking density is generally expressed in ALU ha-1, not the inverse (furthermore it's the definition you use implicitly when evoking a stoking density "double that of the recommended rate" on L105).

L100: Is the value you give (1/16 ALU ha-1) an annual mean or the peak value during the grazing periods? In the first case, what is the peak value?

L105 : I'm surprised by the order of magnitude of the stocking rate (but I'm not familiar with semi arid ecosystems). Do you confirm the number of 0.125 ALU ha-1 for the HG site? It seems very small to me, being rather familiar with stocking rates over 2 ALU ha-1 (but in a totally different ecosystem, of course).

Data processing

No specific comment on the flux computation procedure. It appears correct and well implemented. As mentioned above, I anyway have a problem with the HG site. As the cattle is confined in a paddock and regularly moved from one paddock to another, this would need a specific data treatment (see above).

Uncertainties

There is no standard way to estimate the uncertainties but I think that the procedure followed here takes the most important uncertainty factors into account. I'm thus OK with it.

L204: Shouldn't RE be at the square in the sum ?

L206: Confusion between lower and upper case delta in in Eq 3 and in Eq 1 and L199 and maybe elsewhere. Please harmonize.

Results

Many figures and Tables are redundant. Besides, some key information (the annual budget for each site) does not appear in the Tables. I think that Figure 4 could be skipped as it does not bring more information than Figures 3. This is also the case for Figures 6, 7 and 8. All the information they bring is already given in Figure 5. In addition, I'm not convinced by the interest of giving min, mean and max values of Fc diurnal course (Table 2), daily (Table 3) or monthly (Table 4) cumulative fluxes as these values were not really discussed in the text. Besides this, a synthetic Table providing cumulated values of NEE, GPP and Reco for the different seasons and for the whole year (Something like Table 5 but completed with annual means and GPP and Reco data) would be expected.

The authours use different time scales to describe flux seasonal evolution: data are averaged sometimes on a monthly basis, sometimes on a four season basis and sometimes on a two season basis (dry and wet). This engenders confusion and is also maybe a cause of the table and figure redundancy. I suggest the authors to select only the most relevant time scale and to use it for all flux and meteorological variable descriptions.

The results present mainly chronological evolution of fluxes, which is a little bit scarce. The discussion could much more comprehensive if based on functional relationships (response of fluxes to driving variables) and on their site to site or inter-annual variability.

Flux numbers were often given with two decimals. This is useless and decimals have absolutely no meaning as the uncertainty is of the same order of magnitude than NEE itself. Rounding all flux numbers at the unit is probably more appropriate (and by the way, more readable).

The enumeration of numbers in the text makes its reading quite fastidious. It could most of the time be avoided and focus on the most striking and original points.

Figure 2: Presenting half hourly data of Tair, RH and PPFD makes the figure difficult to read and, especially, does not allow the reader to discern inter-annual differences. I suggest replacing half hourly measurements by day averages and, when relevant, extrema).

L261-285: The interest of the figure 3 is that it provides information on the inter-seasonal, inter-annual and site to site variability of this response. The text should rather emphasize these differences.

Figure 5a: I'm puzzled by the livestock period patterns in the LG: all grazing periods seem to correspond to periods where vegetation is not active. Again I'm not familiar with cattle management in semi arid regions but this appears illogical to me as, at these moments, vegetation should be absent or senescent and thus unpalatable for cattle.

Figure 5b: In the HG, the grazing pattern should highlight the alternation between the two paddocks.

Figures 5: By comparing the flux evolution with climate, it seems that the vegetation activity is not in phase with the solar radiation or with the temperature. This is maybe a specificity of semi arid sites (compared to temperate sites). As the paper is intended for an international audience that is not necessary familiar with South African climate, it could be nice to more clearly show the relation between meteorological variable and flux annual patterns.

L310-316: The text could be made more attractive by avoiding fastidious number enu-merations and focusing on the most striking points. For example the lag between meteorological variables and fluxes evoked above could be better explained here. In addition, the inverted GPP and Reco peaks that appear during the growing season on several years (most clearly on Jan-Feb, Year II) are unusual and call for a comment. How do you explain them?

Figure 9: This figure is the second most relevant one as it clearly highlights the interannual and site to site flux variability. It would be worth adding in each figure shaded areas indicating the cumulated uncertainties of each NEE.

L345-354: This paragraph contains the numbers, presented as the most important of the paper (as they are cited in the abstract). It is thus strange that they are presented in a text and not in a Table (while most of the Tables contain dispensable results). I would add that the number presentation is awkward (no sign difference between sources and sinks, order of the sites changed during the presentation) which makes difficult to the reader to build a budget by himself.

L353: What do represent the numbers after the +/- sign? Are there cumulated uncertainties on a 4 year budget, do they represent interannual variability or are there a mixing of both? This point should also be clarified in the Tables.

Figure 10: A good correlation between NDVI and GPP seasonal evolutions is the least than could be expected and is not very informative. It would be much more interesting to test the ability of NDVI measurements to reproduce inter-site GPP differences (maybe on the two first years) or inter-annual GPP differences (maybe by comparing Years III and IV).

Discussion

I have some difficulties to comment this discussion as I disagree on the point of departure, as said before. To my opinion, the introducing paragraph (L275-278) is incorrect in view of the carbon budgets and the uncertainty analysis presented before. Consequently, I also consider that all the sentences that are based on this assertion are also incorrect. I will not enumerate all of them.

The discussion on the development of unpalatable grasses in the HG site due to past overgrazing and its impact on ecosystem response to drought could anyway be interesting. Unfortunately it does not rely on a quantification of the parcel species composition and, in addition, does not lead to significant flux differences. This point could

maybe be deepened by looking more closely the flux response to SWC and the way it differs between HG and LG during the drought periods (after getting rid of present grazing impact).

L390-393: In what the fact that grasses are stronger competitors for water implies that the studied sites were net sources? The link is not clear at all.

L422: I think that this is an over-interpretation. Several observations challenge it:(i) on the HG site, the NDVI was the lowest on Year I while there was no grazing; (ii) the decrease in NDVI peaks is also observed on the LG site; (iii) a lower plant development on Year IV compared to Year III could also be explained by the drought conditions. I think that, in view of the variability of meteorological conditions and the shortness of the grazing (two years) it is not possible to determine any long term impact of the grazing on the basis of the present measurements.

L424-430: See my comment above on the relation between NDVI and GPP.

L424-430: In addition, this paragraph looks like a catch-all in which several different points of discussion are put without development and without guiding thread.

L448-465: This paragraph is more or less a repetition of observations already made before. It is only descriptive and chronological. Example of questions (among others) I would have liked to see discussed: Precipitations are known to affect both GPP and TER (but at different time scales). How does it happen at your site and does it provoke an increase or a decrease of NEE in the end? As the two sites vegetation compositions are different, can you compare the impact on GPP and Reco dynamics of rain or drought? Are they different?

L473-478: I'm not convinced by the relevance to compare local carbon budgets with a global estimate that could encompass very different ecosystem types.

Conclusions

See my general comments above.

---

## Referee Comment (RC2) · Marc Aubinet (Referee) · 4 Jan 2021

Most of the comment I made in this review are quite basic as they highlight elementary errors (insufficient treatment of the data and over-interpretation of the results, notably). By having often worked with PhD students in their early career, I know that such "youthful errors" are frequent. They are, to my opinion excusable and probably normal for beginner scientists.

However, it is the responsibility of the promoter(s) to correct them and help the PhD student driving his/her analysis and this should have been done during the writing process, before submitting the paper. As a promotor, I would never have authorized

a PhD student or a Post Doc to submit a manuscript in this shape. By reviewing this paper, I had the bad feeling to do the work of the promoter in his/her place.

---

## Referee Comment (RC3) · Anonymous Referee #2 · 11 Jan 2021

Thank you very much for letting me read this very interesting manuscript. CO2 flux sites are too few at the African continent, so your work providing two new sites in Africa is of very high relevance for all of us working within both the Eddy covariance community, and within general climate and Earth system sciences. This manuscript set out to investigate the impact of different grazing regimes on land atmosphere of CO2. And having this very nice data for two adjacent semi-arid savannah site in South Africa is a fantastic opportunity to make some very interesting analysis. This manuscript thereby has great potential in increasing our understanding in carbon cycle dynamics for semi-arid savannah landscapes. However, I have some major concerns regarding the presentation of the manuscript as outlined below.

[Figure]

General comments:

1) As mentioned, the main aim is to study the impact of grazing by comparing two adjacent sites with similar meteorological and hydrological conditions but with different grazing regimes. However, no real analysis comparing the two sites is provided. It is claimed that there is a significant difference. But not uncertainty estimates around either the budgets, or the environmental conditions are provided, and it is hence impossible to see if the differences are significant. I doubt that there is a significant difference between the two sites, given the high variability of the fluxes, and that the fluxes of the two sites still are relatively close to each other. You hypothesize that the HG regime reduces the ecosystem carbon sink potential by altering vegetation cover, decreasing above-ground biomass (AGB) and gross primary production. But no test of this hypothesis is presented. Such an analysis must be provided in order to draw the conclusions presented in the manuscript.

2) Instead of focusing the results on the main aim, that is to study the impact of the grazing on the budgets comparing the sites, the results are basically just one long report of various CO2 flux budgets for different temporal averaging periods. No results of an actual analysis comparing the sites is provided. It is fine to have a section in the beginning of the results describing the hydrological and meteorological conditions as well as a first presentation of the fluxes. However, while reading the results I was continuously waiting for the actual results to start. The main focus of a results section should be to fulfill the aims set out in the introduction. As it is now it is too much focus on reporting flux budgets, and too little on comparing the difference between the sites. I would recommend to streamline the results substantially, and move quite a lot of the currently presented results/figures to an appendix/supplementary information.

3) The Introduction and methods section reads very well, but both the Results and the Discussions must be streamlined with the aim of the paper. The conclusions drawn in the discussions are not firmly based on the presented results (see further comments below).

Minor comments: Include standard deviation of the quantified sink and sources (L21).

If "The two sites differed in soil heterogeneity and characteristics particularly in stone content (soil skeleton >2 mm for the HG site)" (L131). Should this not have a substantial influence on the difference in the $CO_2$ flux budgets between the sites?

(L186-189) Why was it decided to use the night time partitioning method? Is there a strong relationship between $CO_2$ fluxes and temperature? I think in general the respiration-temperature relationship is pretty weak for semi-arid ecosystems. I think the daytime partitioning method is better under these circumstances.

I do not quite understand how the systematic errors were included in the uncertainty estimates. In equation 3, only random and gap filling bias is included? Whereas at L194 it is stated that systematic errors associated with advection, flux divergence and tilt correction, were taken into account. Where in the results is the uncertainty estimates presented and used?

Why use both MOD13Q1 and MYD13Q1? (L215) Would it not be enough with one of the products. What extra info is gained by using both Aqua and Terra time series? How were they combined, given that only one time-series is presented in Fig 2?

The footprint is very short (L260). How was it calculated? There is no description in the methodology.

I would recommend to move the separation of hydrological years to the method section.

A lot of figures and tables present the same results. In the interest of streamlining the manuscript I would recommend to move a lot of presented results to an appendix, and instead focus the results on an analysis comparing the two sites to see if a significant difference between the sites can be seen.

Table 4, What is behind the $\pm$? One standard deviation based on inter-annual variability? Or is it the uncertainty from the uncertainty estimates?

[Figure]

Please include uncertainty around the cumulative fluxes of Figure 9. I would also recommend to skip the final figure covering the full study period, it is not really of importance how they differ over a 4-year period. One extra with the average year for both sites could be interesting, to see if the two sites on average differ from each other.

What was the following conclusion based on: "the two investigated grazing regimes under similar climate, soil conditions and topography have highly influenced plant species composition and vegetation cover leading to implications for their role as potential grazing areas and/or efficient $CO_2$ sinks". I doubt that the vegetation cover between the sites is significantly different (NDVI Fig 2). The fluxes also seem to be very similar at diurnal (Fig 3); seasonal (Fig 5 and 6), and if including the uncertainty, most likely also the inter-annual scale.

(L381) Please include standard deviations around the budgets, to make sure that the sites are significantly different from zero (really being sinks and sources) and from each other.

(L384) How can we tell that there is a difference in Aboveground biomass. The difference in NDVI seems to be minimal, is there any way to test if the difference is significantly higher? How can we tell that it was caused by overgrazing in the past? Could it not be the current grazing as well?

(L385) During most of the resting periods there is no difference between the sites, and the site difference does not seem to be dependent on if it is resting or grazing periods.

I do not understand how a conclusion regarding the impact of the long resting period can be drawn in the discussions. First, there is still grazing going on, so there is no way the impact of the long resting period from the current grazing regime can be separated. Secondly, previously in the manuscript it was stated that the effect of the heavy grazing was still evident and that the grazing that was continued after 2017 warranted that the HG site could be used a heavy grazed site. In this case the long resting period should not have an impact. The heavy grazing is continued from 2017 and onwards. Could

it not rather be so that the heavy grazing increases the $CO_2$ uptake? (Tagesson et al 2016 in reference list). If this now really is the case.

L410 Please explain. I cannot see a statistically significantly higher NEE for HG than for LG in Fig 3. Quite the opposite, I see no significant difference?

(L432) Where is it shown that the inter-annual variability is caused by rainfall/SWC? A start of the growing season with start of the rainfall is no real surprise, that is the general case for semi-arid ecosystems (without dense tree cover). But it is not shown it in any figure; there is no place where a start of the rainy season is linked with the start of the growing season. It is also claimed that the inter-annual budgets are caused by the rainfall variability, but no actual analyze of such a relationship is presented.

L448-L463 This is not a discussion, it is just a long repetition of periods of rainfall and $CO_2$ fluxes. Please do some analysis instead, present the results in the results section and then discuss these results.

The conclusion that "the high ratio of unpalatable vs palatable species made this site less suitable for its current use as sheep pasture" is not a conclusion of the presented results. Please present results that allows to draw such a conclusion. If current conclusions that HG has significantly higher $CO_2$ uptake than LG holds for a statistical test, why is then the conclusion not that the grazing regime of HG is better than LG?

---

## Author Comment (AC1) · 4 Feb 2021

We appreciate the useful suggestions made by the reviewer and would like to thank you for taking the time to review our manuscript. We have carefully gone through the points that were raised. In the following, we provide our responses with blue colour.

**Synthesis**

The paper presents $CO_2$ budgets of semi-arid grasslands/shrubs ecosystems in Karoo, South Africa with a special focus on grazing. Two sites, one considered as submitted to heavy grazing, the other to a lenient grazing, were compared. $CO_2$ fluxes were measured using the eddy covariance technique.

Semi-arid ecosystems and, more generally, African ecosystems are not enough investigated and poorly covered by flux measurements. In addition, as measures are sought to mitigate climate change, especially in the livestock sector, it is of primary importance to better understand and quantify the impact of cattle management on ecosystem carbon balance. I think thus that the topic is of great interest. In addition, it perfectly fits the scope of the journal.

We agree and would like to stress the importance of these datasets. There is an extreme deficit of flux measurements from Africa, which has a major downstream impact on understanding processes in 20% of the global land surface. The context here is very different to other parts of the world where many more measurements are available. As such the observations conducted in this study contribute significantly to improving the availability of measurements.

The methodology used to measure and treat the eddy covariance data seems correct and well implemented. I have however one restriction concerning the HG site that I will detail below.

The general paper shape (quality of writing, figure quality, reference list) is good.

I thus think that the measurements realised here have the potential to make a good paper that would deserve publication.

We thank the reviewer for these positive comments.

However, I have a strong reservation concerning the result presentation and discussion. The results have not been sufficiently addressed, being most of the time limited to a chronological presentation of the fluxes. The analysis is too superficial, premature and based on an over-interpretation of the results, I will detail this below. I thus think that the paper needs a complete reshape and, in view of the importance of the needed changes, I suggest rejection of the paper and I encourage the authors to resubmit it after a complete redesign of their result presentation and analysis.

We understand the reviewer's points (see responses below), however we feel that we will be able to properly address these comments and modify the manuscript according to a major review.

**General comments**

The two main theses in this paper are that (i) site to site differences between carbon budgets is mainly determined by the current and past management and that (ii) their interannual variability is linked to soil humidity.

The first point is announced from the beginning and repeated several times in the paper but, unfortunately, never really supported by the measurements:

First, the budget over four years yields to 82.11 and -36.43 gCm-2 (corresponding to annual budgets of 20.5 and -9.1 gCm-2yr-1) for LG and HG, respectively. In view of the uncertainties due to the measurement technique and of the interannual variability, I doubt that these budgets are significantly different and, moreover, that each budget is significantly different from zero. The interpretation (which is central to the paper) that one site behaves as a source and the other as a sink is thus abusive. It is most probable that none of the site significantly depart from equilibrium.

As we used an honest and transparent way of reporting uncertainties, it is in our opinion mostly a discussion whether the two sites can be called "different" or not and then of course whether the conclusions we draw hold. A lot can be done by streamlining the results and using a more appropriate wording. In general, results - even if not significantly different from zero - deserve publication, they still are a result. Holding these results back would put a bias into the scientific literature when only large fluxes would be shown.

In the revised version, we plan to include statistical tests to demonstrate whether or not datasets are significantly different. Analyses will cover differences between sites, differences between growing seasons of sites, differences between years, probably binning year 1 and 2 vs. year 3 and 4 to take different grazing into account. Our conclusions will be aligned accordingly.

Secondly, even if a significant site to site difference had to be found (maybe on the two first years), it could be attributed as well to present as to past management. Even if a possible impact of present management is evoked at some place, the reader is leaded to think that the past management is the most important factor.

The studied sites are part of long-term grazing experiments at a local agricultural research institute (Grootfontein Agricultural Development Institute, GADI, Middelburg, Eastern Cape, South Africa), and the impacts of the different grazing systems on vegetation are well known and researched. The HG site was grazed for a long period (1988-2007) with stocking rates double of the recommended rates. This severe

treatment extirpated nearly all palatable species and nearly all dwarf shrubs, and as a result, the system is dominated by unpalatable shrubs. We will add detail on the differences between the sites, and also formulate the text differently, where the impacts about past and present management are not currently clear.

Indeed, in the material and method, a description of present management is missing and we have to wait for the result presentation (Figure 5) to realize that present management differed completely, not only between the two sites but also between the beginning and the end of measurement period in one site, the so-called "heavy grazed" site being not grazed at all during the two first years (!!). As a result the impact of presented grazing on the flux has been completely overlooked, which biases the conclusions and the discussion.

We agree with the reviewer that details of the present and past management were lacking. We will 1) re-name the current HG site as "Experimental Site (EG)" to emphasize the experimental management, rather than continuous high grazing, and 2) add information about the current and past livestock management into the materials and methods section.

The analysis of the long term impact of overgrazing based on the differences between parcel species composition, with a greater abundance of unpalatable grasses in the HG site, is promising. Unfortunately, the analysis is only qualitative and does not rely on a quantitative inventory of species on each site.

Quantitative species inventories are conducted by the agricultural development institute GADI and available (however mainly unpublished; as referenced), and it was not in the focus of this study to conduct further inventories. We will add a table based on the inventories, listing the most common species in both sites, to help the reader get an idea of the impacts of overgrazing on vegetation.

Concerning the second point (response to humidity), it would be relevant to attribute the interannual variability to precipitation and soil moisture differences. This aspect is however insufficiently developed to my opinion and results presenting the responses of NEE, GPP and Reco to SWC should be deepened.

We appreciate this suggestion. We will follow the reviewer's advice and develop this aspect with an additional graph that will present the NEE, Reco and GPP responses to the SWC.

Finally, I have some reservations as regards the measurement methodology on the HG site for the grazed years: the authors announced that the site was divided in two paddocks and that the cattle were regularly moved from one paddock to another. I suppose thus that CO2 fluxes should differ if they come from the occupied or from the empty paddock due to cattle respiration but also to vegetation consumption. However, no information about the paddock positioning with respect to the measurement tower or about cattle moving chronology was given; no flux discrimination between the empty

and the occupied paddock was tempted. In these conditions, I suspect that the fluxes should be strongly affected by footprint variations. Discriminating fluxes between the paddocks, if possible, could provide information on the impact of present grazing on fluxes.

We will further add detail and background to justify our assumptions, and revise the conclusions in relevant parts. We agree that the grazing systems need to be described in more detail. Here, it is relevant to note that the grazing systems in the Karoo are extremely low-intensity and as such, can not be compared to the intensive grazing in temperate systems. We believe that this is the reason for much of the apparent misunderstanding regarding the reporting of the detail on the systems, as in the comments below and from Reviewer #2.

I recommend the authors reading papers by Jerôme et al., AEE 194, 7-16, 2014; Felber et al., BG 13, 2959-2969, 2016; Gourlez de la Motte et al., AFM 268, 156-168, 2019, and references herein to better address the difficult question of highlighting the grazing impact on carbon balance. 3

We are aware of these studies, however, the systems presented are in no way comparable with the here presented Karoo systems. Beside the fact that the Karoo systems are grazed with sheep, the animal density is less than a tenth of the European systems described in the listed papers, thereby leaving it questionable whether animal positioning and outgassing has an impact on the $CO_2$ fluxes detected by the tower. Unfortunately, exact information on animal positioning is not recorded in these systems, but we can certainly add information on paddock location.

**More specific comments**

**Abstract**

L21: As already said above, in view of the uncertainties and of the interannual variability, I think that the difference between the two sites is not significant (and a source of 82 gC m-2 over 4 years is by no means "considerable" as the uncertainty on the annual sequestration is about 50 gC m-2yr-1). I found thus misleading to present one site as a source and the second as a sink. Both sites do probably not depart significantly from equilibrium. As this assertion is central to the paper, it questions all the discussion and conclusions.

We understand the reviewer's point, however we argue that the assertion does not question the discussion and conclusions, and that the paper has strong value even if we change the interpretation and wording here. In a highly variable system, where the main variability is controlled by rainfall dynamics, it is difficult to tease out the impact of management. We will streamline the results by using a more appropriate wording, and adjust discussion and conclusions accordingly.

**Introduction**

L78-80, Independently of the fact that I disagree with the sentence, an introduction should finish with a presentation of the objectives of the papers in the form of scientific questions, not with a result.

This is probably a misunderstanding as the sentence is related to the presentation of the hypotheses, and explains how we addressed the objectives (and is not a result per se).

**Sites description**

A section describing the cattle management during the measurement period (stocking rate evolution, chronology, position of the paddocks) is missing. The fact that the so-called "heavy grazed" site is not grazed at all during the first two years should appear more clearly (and the name of the site should be changed as it is strongly misleading!).

We will add a section describing the cattle management during the measurement period. The management of the re-named Experimental Grazing site (previously HG) will be explained earlier on in the manuscript. We will better elaborate the expectations related to the impacts of past and current management in the hypotheses, results, and conclusions.

What are the parcel sizes?

The sizes of the paddocks are 200 x 250 m (LG) and 550 x 200 m (HG); we will add further information regarding the size and positioning of the parcels within the text.

L100: The stocking density is generally expressed in ALU ha-1, not the inverse (furthermore it's the definition you use implicitly when evoking a stoking density "double that of the recommended rate" on L105).

We will replace ha $AU^{-1}$ with ALU $ha^{-1}$.

L100: Is the value you give (1/16 ALU ha-1) an annual mean or the peak value during the grazing periods? In the first case, what is the peak value?

In all cases the stocking rates are long-term stocking rates.

L105 : I'm surprised by the order of magnitude of the stocking rate (but I'm not familiar with semi arid ecosystems). Do you confirm the number of 0.125 ALU ha-1 for the HG site? It seems very small to me, being rather familiar with stocking rates over 2 ALU ha-1 (but in a totally different ecosystem, of course).

The stocking rates in the Karoo semi-arid ecosystems are very low, compared to temperate grazing systems. The stocking rate at the HG site was 0.220 ALU $ha^{-1}$.

**Data processing**

No specific comment on the flux computation procedure. It appears correct and well implemented. As mentioned above, I anyway have a problem with the HG site. As the

cattle is confined in a paddock and regularly moved from one paddock to another, this would need a specific data treatment (see above).

See comment above.

**Uncertainties**

There is no standard way to estimate the uncertainties but I think that the procedure followed here takes the most important uncertainty factors into account. I'm thus OK with it.

L204: Shouldn't RE be at the square in the sum ?

Equation (2) was not correctly displayed. We modify it to:

$$\varepsilon ASum = \sqrt{\sum_N RE_i^2}$$

L206: Confusion between lower and upper case delta in in Eq 3 and in Eq 1 and L199 and maybe elsewhere. Please harmonize.

We will use lower case delta throughout the manuscript.

**Results**

Many figures and Tables are redundant. Besides, some key information (the annual budget for each site) does not appear in the Tables. I think that Figure 4 could be skipped as it does not bring more information than Figures 3. This is also the case for Figures 6, 7 and 8. All the information they bring is already given in Figure 5. In addition, I'm not convinced by the interest of giving min, mean and max values of Fc diurnal course (Table 2), daily (Table 3) or monthly (Table 4) cumulative fluxes as these values were not really discussed in the text. Besides this, a synthetic Table providing cumulated values of NEE, GPP and Reco for the different seasons and for the whole year (Something like Table 5 but completed with annual means and GPP and Reco data) would be expected.

We will move Figs. 4, 7 and 8 and Tables 2, 3 and 4 to the appendix. An Additional table with cumulative values of NEE, GPP and Reco (seasonal and annual) will be added. However, we decided to leave Figure 6 in the main text because it helps to compare the interannual, seasonal and daily exchange of carbon fluxes as well as the length and strength of carbon uptake during the day, which differed each year (Fig. 3 represents mean values).

The authours use different time scales to describe flux seasonal evolution: data are averaged sometimes on a monthly basis, sometimes on a four season basis and sometimes on a two season basis (dry and wet). This engenders confusion and is also

maybe a cause of the table and figure redundancy. I suggest the authors to select only the most relevant time scale and to use it for all flux and meteorological variable descriptions.

The different time scales were used in order to emphasize the differences between sites at the different (daily, monthly, seasonal and annual) scales. The most relevant time scale was selected in each individual case.

The results present mainly chronological evolution of fluxes, which is a little bit scarce. The discussion could much more comprehensive if based on functional relationships (response of fluxes to driving variables) and on their site to site or inter-annual variability.

We will develop further discussion based on the functional relationship between SWC and fluxes and based on a clearer comparison of differences between sites and seasons.

Flux numbers were often given with two decimals. This is useless and decimals have absolutely no meaning as the uncertainty is of the same order of magnitude than NEE itself. Rounding all flux numbers at the unit is probably more appropriate (and by the way, more readable).

We will round all the flux numbers.

The enumeration of numbers in the text makes its reading quite fastidious. It could most of the time be avoided and focus on the most striking and original points.

We will undergo a revision of the text to reduce the numbers, where relevant, and to better focus on the most striking points.

Figure 2: Presenting half hourly data of Tair, RH and PPFD makes the figure difficult to read and, especially, does not allow the reader to discern inter-annual differences. I suggest replacing half hourly measurements by day averages and, when relevant, extrema).

We will modify Figure 2 following the reviewer's suggestion (using daily mean values) while half-hourly data will be transparently shown in the background.

L261-285: The interest of the figure 3 is that it provides information on the interseasonal, inter-annual and site to site variability of this response. The text should rather emphasize these differences.

We will adjust the text and emphasize these differences.

Figure 5a: I'm puzzled by the livestock period patterns in the LG: all grazing periods seem to correspond to periods where vegetation is not active. Again I'm not familiar with cattle management in semi arid regions but this appears illogical to me as, at

these moments, vegetation should be absent or senescent and thus unpalatable for cattle.

There are two factors that explain the livestock period patterns; First, vegetation biomass is almost never absent in the studied systems. Shrubs retain much of their size, and grass tufts are partially grazed. Animals will move to the next paddock long before all vegetation is removed. Second, non-growing vegetation retains its quality well (almost like standing hay) and remains palatable to animals.

Figure 5b: In the HG, the grazing pattern should highlight the alternation between the two paddocks.

We will provide additional information about the paddock system (however, also see previous responses regarding the grazing system).

Figures 5: By comparing the flux evolution with climate, it seems that the vegetation activity is not in phase with the solar radiation or with the temperature. This is maybe a specificity of semi arid sites (compared to temperate sites). As the paper is intended for an international audience that is not necessary familiar with South African climate, it could be nice to more clearly show the relation between meteorological variable and flux annual patterns.

We will provide additional information about seasonality in South Africa in this context (see e.g. du Toit and O'Connor., 2014). These ecosystems are highly driven by water availability, while temperature and radiation are not necessarily the main drivers. See also the references used in the introduction (L56-59).

L310-316: The text could be made more attractive by avoiding fastidious number enumerations and focusing on the most striking points. For example the lag between meteorological variables and fluxes evoked above could be better explained here. In addition, the inverted GPP and Reco peaks that appear during the growing season on several years (most clearly on Jan-Feb, Year II) are unusual and call for a comment. How do you explain them?

It remains unclear to us what the reviewer means with "inverted GPP and Reco peaks". In most of the years there were at least 2 GPP and Reco peaks in the growing season. These peaks were in phase, not necessarily matching in amplitude. These peaks corresponded with rain events showing the immediate response of the systems to water during the growing season.

Figure 9: This figure is the second most relevant one as it clearly highlights the inter-annual and site to site flux variability. It would be worth adding in each figure shaded areas indicating the cumulated uncertainties of each NEE.

Good point. We will add in each year shaded areas indicating the cumulative uncertainties.

L345-354: This paragraph contains the numbers, presented as the most important of the paper (as they are cited in the abstract). It is thus strange that they are presented in a text and not in a Table (while most of the Tables contain dispensable results). I would add that the number presentation is awkward (no sign difference between sources and sinks, order of the sites changed during the presentation) which makes difficult to the reader to build a budget by himself.

We thank the reader for the suggestion, and will create a Table with annual cumulative fluxes. Also, we will make sure that the differences between sources and sinks are easy to read. Please note that in cases where we explicitly talk about sources or sinks, no signs can be used, for example, what would be a sink of -31 g C m$^{-2}$ yr$^{-1}$? That would confuse the reader even more.

L353: What do represent the numbers after the +/- sign? Are there cumulated uncertainties on a 4 year budget, do they represent interannual variability or are there a mixing of both? This point should also be clarified in the Tables.

The +/- sign on the L353 represents the average value of the four-year annual uncertainties. We will clarify this in the text.

Figure 10: A good correlation between NDVI and GPP seasonal evolutions is the least than could be expected and is not very informative. It would be much more interesting to test the ability of NDVI measurements to reproduce inter-site GPP differences (maybe on the two first years) or inter-annual GPP differences (maybe by comparing Years III and IV).

We agree that the information on NDVI should be further explored. As the inter-site differences are relatively low, see Figure 2e, we will rather focus on the inter-annual variability and its possible relation to the annual total CO2 budget.

**Discussion**

I have some difficulties to comment this discussion as I disagree on the point of departure, as said before. To my opinion, the introducing paragraph (L275-278) is incorrect in view of the carbon budgets and the uncertainty analysis presented before. Consequently, I also consider that all the sentences that are based on this assertion are also incorrect. I will not enumerate all of them.

The discussion on the development of unpalatable grasses in the HG site due to past overgrazing and its impact on ecosystem response to drought could anyway be interesting. Unfortunately it does not rely on a quantification of the parcel species composition and, in addition, does not lead to significant flux differences. This point could maybe be deepened by looking more closely the flux response to SWC and the way it differs between HG and LG during the drought periods (after getting rid of present grazing impact).

The EC towers are placed on experimental grazing sites of a local agricultural research institute. This means that quantitative species inventories are available, and it was not in the focus of this study to conduct further inventories. We will, however, add detail on the vegetation composition of the two sites, including a table listing the most abundant species of both sites. We will also deepen the discussion following flux response to SWC. We will for example investigate whether the relationship changes simply over time, i.e. after the start of the rainy season, or whether the rainfall intensity plays a role in determining the flux response to SWC.

L390-393: In what the fact that grasses are stronger competitors for water implies that the studied sites were net sources? The link is not clear at all.

"It implies that grasses are stronger competitors for water, especially in water-limited ecosystems with pulsed precipitation." This sentence suggests that unpalatable grasses - due to the abundant fine-root biomass in the upper soil layer - are able to absorb water faster compared to shrubs with deep-root systems that use water in deeper soil layers. Thus, during a small rain event in the dry season, unpalatable grasses will germinate, while water will not reach the shrubs' root system due to evaporation.

L422: I think that this is an over-interpretation. Several observations challenge it:(i) on the HG site, the NDVI was the lowest on Year I while there was no grazing; (ii) the decrease in NDVI peaks is also observed on the LG site; (iii) a lower plant development on Year IV compared to Year III could also be explained by the drought conditions. I think that, in view of the variability of meteorological conditions and the shortness of the grazing (two years) it is not possible to determine any long term impact of the grazing on the basis of the present measurements.

We understand the reviewer's point. We will reformulate this paragraph and review the interpretation of NDVI. Furthermore, we will conduct a statistical difference test for the NDVI data between sites, Years I-II & III-IV and the growing & dry seasons. We will also add a bar plot to visually compare NDVI between sites (Year I & Year III and seasons (Dry & Wet).

L424-430: See my comment above on the relation between NDVI and GPP.

L424-430: In addition, this paragraph looks like a catch-all in which several different points of discussion are put without development and without guiding thread.

See our comment above.

L448-465: This paragraph is more or less a repetition of observations already made before. It is only descriptive and chronological. Example of questions (among others) I would have liked to see discussed: Precipitations are known to affect both GPP and TER (but at different time scales). How does it happen at your site and does it provoke an increase or a decrease of NEE in the end? As the two sites vegetation compositions

are different, can you compare the impact on GPP and Reco dynamics of rain or drought? Are they different?

The significance of this paragraph is that it more descriptively explains the difference between the first and second year (when the annual precipitation was almost the same for both years) and why our fluxes (especially annual cumulative NEE) acted differently in the Year I and II with same annual precipitation. Thus, this explains the importance of the distribution of precipitation during the year in such water-limited ecosystems.

L473-478: I'm not convinced by the relevance to compare local carbon budgets with a global estimate that could encompass very different ecosystem types.

There is generally an extreme deficit of flux measurements from Africa, which has a major downstream impact on understanding processes in 20% of the surface of the global land. It may have important implications on the broader scale (African continent) as such semi-arid ecosystems are understudied.

**Conclusions**

See my general comments above.

---

## Author Comment (AC3) · 4 Feb 2021

We would like to thank the reviewer for her/his evaluation of our manuscript and for the suggestions that were given. We have carefully gone through the points that were raised. In the following, we provide a response in blue colour.

**Anonymous Referee #2**

**Received and published: 11 January 202**

Thank you very much for letting me read this very interesting manuscript. CO2 flux sites are too few at the African continent, so your work providing two new sites in Africa is of very high relevance for all of us working within both the Eddy covariance community, and within general climate and Earth system sciences. This manuscript set out to investigate the impact of different grazing regimes on land atmosphere of CO2. And having this very nice data for two adjacent semi-arid savannah site in South Africa is a fantastic opportunity to make some very interesting analysis. This manuscript thereby has great potential in increasing our understanding in carbon cycle dynamics for semi-arid savannah landscapes. However, I have some major concerns regarding the presentation of the manuscript as outlined below.

**General comments:**

1) As mentioned, the main aim is to study the impact of grazing by comparing two adjacent sites with similar meteorological and hydrological conditions but with different grazing regimes. However, no real analysis comparing the two sites is provided.

The manuscript will be reorganised with a few new figures, but also some of the key figures will be kept. These are Figs. 2, 3, 5, and 9. They already provide a direct comparison of either environmental conditions or fluxes. The new figure will further stress a direct comparison NDVI between sites (Year I & Year II) and seasons (Dry & Wet).

It is claimed that there is a significant difference. But not uncertainty estimates around either the budgets, or the environmental conditions are provided, and it is hence impossible to see if the differences are significant. I doubt that there is a significant difference between the two sites, given the high variability of the fluxes, and that the fluxes of the two sites still are relatively close to each other.

Uncertainty estimates were made for cumulative monthly, seasonal and annual fluxes. We used an honest and transparent way of reporting uncertainties, it is in our opinion mostly a discussion whether the two sites can be called "different" or not and then of course whether the conclusions we draw hold. Nevertheless, we will add further analysis investigating the differences between two sites. See our detailed response to major point 2.

You hypothesize that the HG regime reduces the ecosystem carbon sink potential by altering vegetation cover, decreasing above-ground biomass (AGB) and gross primary production. But no test of this hypothesis is presented. Such an analysis must be provided in order to draw the conclusions presented in the manuscript.

We understand the reviewer's point. We will review the interpretation of NDVI by doing a statistical difference test for the NDVI data between sites, Years I-II & III-IV and the growing & dry seasons. Also, we will add bar plots to compare NDVI between sites (Year I & Year II) and seasons (Dry & Wet).

2) Instead of focusing the results on the main aim, that is to study the impact of the grazing on the budgets comparing the sites, the results are basically just one long report of various CO2 flux budgets for different temporal averaging periods. No results of an actual analysis comparing the sites is provided. It is fine to have a section in the beginning of the results describing the hydrological and meteorological conditions as well as a first presentation of the fluxes. However, while reading the results I was continuously waiting for the actual results to start. The main focus of a results section should be to fulfill the aims set out in the introduction. As it is now it is too much focus on reporting flux budgets, and to o little on comparing the difference between the sites. I would recommend to streamline the results substantially, and move quite a lot of the currently presented results/figures to an appendix/supplementary information.

We would like to point the reviewer to Figs. 2, 3, 5, and 9, which already provide a direct comparison of either environmental conditions or fluxes. In the revised version of the manuscript, we plan to include statistical tests to demonstrate whether or not datasets are significantly different. Analyses will cover differences between sites, differences between growing seasons of sites, differences between years, probably binning year 1 and 2 vs. year 3 and 4 to take different grazing into account. Our conclusions will be aligned accordingly. We will also develop an aspect of the relationship between the water availability and fluxes by introducing additional graphs (SWC/P & fluxes).

3) The Introduction and methods section reads very well, but both the Results and the Discussions must be streamlined with the aim of the paper. The conclusions drawn in the discussions are not firmly based on the presented results (see further comments below).

We will perform further statistical tests, introduce two new figures (see our comments above), insert an additional table (with annual cumulative NEE, GPP and Reco) and move Figs. 4, 7, 8 and Tables 2, 3, 4 to the appendix. We will further revise and streamline the results, discussions and conclusions in relevant parts.

**Minor comments:**

Include standard deviation of the quantified sink and sources (L21).

We will include the standard deviation.

If "The two sites differed in soil heterogeneity and characteristics particularly in stone content (soil skeleton >2 mm for the HG site)" (L131). Should this not have a substantial influence on the difference in the CO2 flux budgets between the sites?

There was indeed a slight difference in the soil between the sites with unfortunately no detailed soil surveys available. We assume, however, that the small difference in soil characteristics may not play a significant role in driving differences in $CO_2$ fluxes between the two sites. Inter-site differences were only observed in year I and II. In these years, grazing intensity at each site remained the same, i.e. low at LG site, resting at HG site, and amount of rainfall was more

or less the same. Thus, we attribute the inter-site differences to grazing and the inter-annual variability between year I and II to different rainfall distribution. We will further clarify this in the manuscript. Soil texture may surely have an influence on physical components like diffusion of $CO_2$, but we don't think that in these soil types it plays a major role as it can be seen in our results.

(L186-189) Why was it decided to use the night time partitioning method? Is there a strong relationship between CO2 fluxes and temperature? I think in general the respiration-temperature relationship is pretty weak for semi-arid ecosystems. I think the daytime partitioning method is better under these circumstances.

We used nighttime partitioning because it has the advantage that GPP+Reco = NEE, which is not the case for daytime partitioning, where GPP may become negative in single cases due to uncertainty in the fit. However, we see the reviewer's point and logical reason behind it. We will follow his/her advice and use the daytime method instead. Table 3 and Figure 5 will be updated accordingly.

I do not quite understand how the systematic errors were included in the uncertainty estimates. In equation 3, only random and gap filling bias is included? Whereas at L194 it is stated that systematic errors associated with advection, flux divergence and tilt correction, were taken into account. Where in the results is the uncertainty estimates presented and used?

The approach by Finkelstein and Sims. (2001) of estimating random errors was applied in this study. Also, bias errors from gap-filling of EC data were considered. We will modify the L194 accordingly. Lucas-Moffat et al. (2018) and Moffat et al. (2007) describe the equations to calculate bias and random errors, which are then summed up to give a measure of uncertainty of annual sums. The uncertainty estimates were used after the ± sign for cumulative NEE.

Why use both MOD13Q1 and MYD13Q1? (L215) Would it not be enough with one of the products. What extra info is gained by using both Aqua and Terra time series? How were they combined, given that only one time-series is presented in Fig 2.

We thank the reviewer for this comment. The use of both Terra and Aqua data would have enabled us to increase the temporal resolution of our NDVI time series from 16 to 8 days. However, even though it was originally planned to incorporate both the MOD13Q1 (Terra) and MYD13Q1 (Aqua) products, we ended up analyzing the NDVI from Terra only. Hence, our data set description from L208 to L219 was not entirely correct and we will revise it accordingly. It now only mentions the Terra NDVI product (MOD13Q1) which we used for our investigation.

The footprint is very short (L260). How was it calculated? There is no description in the methodology.

The tower height is just 3 m. The footprint estimation was performed according to the "simple footprint parameterization" described in Kljun et al. (2004). We will add this description to the methodology section.

I would recommend to move the separation of hydrological years to the method section.

The division of hydrological years was written where it was used and discussed for the first time (Section 3.1), for the purpose of facilitating the reading and interpretation of the results.

A lot of figures and tables present the same results. In the interest of streamlining the manuscript I would recommend to move a lot of presented results to an appendix, and instead focus the results on an analysis comparing the two sites to see if a significant difference between the sites can be seen.

We thank the reviewer for this suggestion and will move Figs. 4, 7,8 and Tables 2, 3, 4 to the appendix.

Table 4, What is behind the ±? One standard deviation based on inter-annual variability? Or is it the uncertainty from the uncertainty estimates?

The sign ± represents uncertainty based on the description in Section 2.3.4.

Please include uncertainty around the cumulative fluxes of Figure 9. I would also recommend to skip the final figure covering the full study period, it is not really of importance how they differ over a 4-year period. One extra with the average year for both sites could be interesting, to see if the two sites on average differ from each other.

We thank the reviewer for making this point. We will include the uncertainties in Fig. 9 and add an extra one with four-year average values. However, we think it is useful to keep four years of cumulative NEE to more clearly describe the overall picture.

What was the following conclusion based on: "the two investigated grazing regimes under similar climate, soil conditions and topography have highly influenced plant species composition and vegetation cover leading to implications for their role as potential grazing areas and/or efficient CO2 sinks". I doubt that the vegetation cover between the sites is significantly different (NDVI Fig 2). The fluxes also seem to be very similar at diurnal (Fig 3); seasonal (Fig 5 and 6), and if including the uncertainty, most likely also the inter-annual scale.

We will add estimation of the plant species coverage (%) in order to emphasize differences in species distribution between sites. The plant cover varies a lot over time as is normal with semi-arid systems. Variation in cover is mainly due to germination and growth of annuals, and secondarily to growth of perennials, especially grasses. The species assemblages are known to be affected by long-term grazing regime.

(L381) Please include standard deviations around the budgets, to make sure that the sites are significantly different from zero (really being sinks and sources) and from each other.

We will add uncertainty estimations here .

(L384) How can we tell that there is a difference in Aboveground biomass. The difference in NDVI seems to be minimal, is there any way to test if the difference is significantly higher? How can we tell that it was caused by overgrazing in the past? Could it not be the current grazing as well?

Quantitative species inventories are conducted by the agricultural research institute GADI and available (however mainly unpublished; as referenced), and it was not in the focus of this study to conduct further inventories. We will add a table based on the inventories, listing the most common species in both sites, to help the reader get an idea of the impacts of overgrazing on vegetation. We will modify L 384 to emphasize differences in plant species between sites.

(L385) During most of the resting periods there is no difference between the sites, and the site difference does not seem to be dependent on if it is resting or grazing periods.

I do not understand how a conclusion regarding the impact of the long resting period can be drawn in the discussions. First, there is still grazing going on, so there is no way the impact of the long resting period from the current grazing regime can be separated. Secondly, previously in the manuscript it was stated that the effect of the heavy grazing was still evident and that the grazing that was continued after 2017 warranted that the HG site could be used a heavy grazed site. In this case the long resting period should not have an impact. The heavy grazing is continued from 2017 and onwards. Could it not rather be so that the heavy grazing increases the CO2 uptake? (Tagesson et al 2016 in reference list). If this now really is the case.

The Heavy Grazing (HG) site was grazed by Dorper sheep using a 2-paddock rotational grazing system (120 days grazing followed by 120 days rest) at stocking rates approximately double that of the recommended rate as part of an experimental trial from 1988 to 2007. The site was ungrazed 2008–2017 but did not recover (palatable species did not come back after resting period). The Dorpers were reintroduced at a similar stocking rate in July 2017 (we have 4 years of measurement (Nov 2015 - Nov 2019)). In conclusion, we said that a long resting period, along with the transition of species from palatable shrubs and grasses to unpalatable grasses, affects carbon fluxes. We cannot conclude that heavy grazing increases $CO_2$ consumption. It can be said that species composition at the HG sites has altered and has been unfavorable for Dorper grazing due to overgrazing in the past. Thus the HG site is considered agriculturally degraded. In the same time, there has been a shift to an increased abundance of unpalatable drought-resistant grass species, favorable for carbon sequestration in such water-limited ecosystems.

L410 Please explain. I cannot see a statistically significantly higher NEE for HG than for LG in Fig 3. Quite the opposite, I see no significant difference?

We meant that the HG site had higher carbon sequestration rates compared to the LG site. We rephrase this sentence to make it clear.

(L432) Where is it shown that the inter-annual variability is caused by rainfall/SWC? A start of the growing season with start of the rainfall is no real surprise, that is the general case for semi-arid ecosystems (without dense tree cover). But it is not shown it in any figure; there is no place where a start of the rainy season is linked with the start of the growing season. It is also claimed that the inter-annual budgets are caused by the rainfall variability, but no actual analyze of such a relationship is presented.

We will develop this aspect further and add an additional graph that will present the NEE, Reco and GPP responses to the SWC.

L448-L463 This is not a discussion, it is just a long repetition of periods of rainfall and CO2 fluxes. Please do some analysis instead, present the results in the results section and then discuss these results.

We would disagree on the positioning of this paragraph in the discussion section; it more descriptively explains the difference between the first and second year (when the annual precipitation was almost the same for both years) and why our fluxes (especially annual cumulative NEE) acted differently in the Year I and II with same water availability. Thus, this

explains the importance of the distribution of precipitation during the year in such water-limited ecosystems.

- The conclusion that "the high ratio of unpalatable vs palatable species made this site less suitable for its current use as sheep pasture" is not a conclusion of the presented results. Please present results that allows to draw such a conclusion.If current conclusions that HG has significantly higher CO2 uptake than LG holds for a statistical test, why is the n the conclusion not that the grazing regime of HG is better than LG?

The first sentence is derived from prior knowledge on studies conducted at the sites, as presented under the material and methods section. The reason we present it here is that it puts the $CO_2$ uptake finding in a more interesting light, and thus, is necessary background. We concluded that, unexpectedly, the site with continuous heavy grazing after a long resting period was more efficient as a $CO_2$ sink (due to transition to unpalatable grasses, which may be better able to compete for water). However, at the same time the value of the HG site for sheep grazing is reduced. Previous studies indicate relatively slow recovery from grazing: Seymour et al. (2010) reported that 20 years of recovery period in the Karoo degraded ecosystems restored grazing potential, while not returning all palatable species. As our study is conducted on sites where livestock grazing is conducted and studied, we believe that this provides an interesting additional angle to the interpretation of our results.